

# Biomass burning emissions disturbances on the isoprene oxidation in a tropical forest

Fernando C. Santos[1], Karla M. Longo[2], Alex B. Guenther[3], Saewung Kim[3], Dasa Gu[3], Dave E. Oram[4], Grant L. Forster[4], James Lee[5], James R. Hopkins[5], Joel F. Brito[6] and Saulo R. Freitas[7]

[1] Earth System Science Center, National Institute for Space Research, São José dos Campos, SP, Brazil
[2] Universities Space Research Association/Goddard Earth Sciences Technology and Research, NASA Goddard Space Flight Center, Greenbelt, MD, USA
[3] Department of Earth System Science, University of California, Irvine, CA, USA
[4] National Centre for Atmospheric Science, School of Environmental Sciences, University of East Anglia, Norwich, UK
[5] National Centre for Atmospheric Science, Department of Chemistry, University of York, York, UK
[6] University of São Paulo, São Paulo, SP, Brazil
[7] Universities Space Research Association/Goddard Earth Sciences Technology and Research, NASA Goddard Space Flight Center, Greenbelt, USA

*Correspondence to*: Fernando C. dos Santos (fernando.santos@inpe.br)

**Abstract.** We present a characterization of the chemical composition of the atmosphere of the Brazilian Amazon rainforest based on trace gases measurements carried out during the South American Biomass Burning Analysis (SAMBBA) airborne experiment in September 2012. We analyzed the observations of primary biomass burning emission tracers, i.e., carbon monoxide (CO) and nitrogen oxides ($NO_x$), ozone ($O_3$), isoprene, and its main oxidation products, methyl vinyl ketone (MVK), methacrolein (MACR), and hydroxyhydroperoxides (ISOPOOH). The focus of SAMBBA was primarily on biomass burning emissions, but there were also several flights in areas of the Amazon forest not directly affected by biomass burning, revealing a background with a signature of biomass burning in the chemical composition due to long-range transport of biomass burning tracers from both Africa and the eastern part of Amazonia. We used the [MVK+MACR+ ISOPOOH]/[Isoprene] ratio and the hydroxyl radical (OH) indirect calculation to assess the oxidative capacity of the Amazon forest atmosphere. We compared the background regions (CO<150 ppbv), fresh and aged smoke plumes classified according to their photochemical age ($[O_3]/[CO]$), to evaluate the impact of biomass burning emissions in the oxidative capacity of the Amazon forest atmosphere. We observed that biomass burning emissions disturb the isoprene oxidation reactions, especially for fresh plumes ([MVK+MACR+ISOPOOH]/[isoprene] = 7). The oxidation of isoprene is higher in fresh smoke plumes at lower altitudes (~ 500 m) than in aged smoke plumes, anticipating near the surface a complex chain of oxidation reactions, which may be related to the secondary organic aerosols (SOA) formation. We proposed a refinement of the OH calculation based on the sequential reaction model, which considers vertical and horizontal transport for both biomass burning regimes and background environment. Our approach for the [OH] estimation resulted in values of the same order of magnitude of a recent observation in the Amazon rainforest [OH] $\cong 10^6$ (molecules $cm^{-3}$). During the fresh



plume regime, the vertical profile of [OH] and the [MVK+MACR+ISOPOOH]/[isoprene] ratio showed an evidence of an increase of the oxidizing power in the transition from PBL to cloud layer (1,000 – 1,500 m). These high values of [OH] ($1.5 \times 10^6$ molecules $cm^{-3}$) and [MVK+MACR+ISOPOOH]/[isoprene] (7.5) indicate a significant change above and inside the cloud decks due to cloud edge effects on photolysis rates, which have a major impact on OH production rates.

## 1 Introduction

Terrestrial vegetation emits to the atmosphere a significant amount of biogenic volatile organic compounds (BVOCs), corresponding to 1,150 Tg Carbon per year. The most abundant BVOC is isoprene ($C_5H_8$), with an annual global emission ranging from 440 to 660 Tg Carbon per year, depending on driving variables such as temperature, solar radiation, leaf area index, and plant functional type (Guenther et al., 2006). In contrast, the global emission rate of anthropogenic volatile organic compounds (AVOCs) is around 145 Tg Carbon per year (Janssens-Maenhout et al., 2015). The atmosphere has a natural mechanism to

balance the VOCs emitted by anthropic and biogenic sources via a complex chain of oxidation reactions, not yet fully understood, followed by the deposition of sub-products, mostly secondary organic aerosols (SOA) (Prinn, 2014). These oxidation reactions occur mainly through the hydroxyl free radical (OH), which has been often used to express the oxidative capacity of the atmosphere. Therefore, the VOCs play an important role in the atmospheric chemistry, influencing the concentrations of ozone ($O_3$) and OH as well as the conversion rates of nitrogen oxides ($NO_x=NO+NO_2$). The VOCs also affect the atmospheric secondary

organic aerosols (SOA), which alter the solar radiation budget and cloud droplet nucleation. Moist regions with high availability of solar radiation, such as the Amazon region, affect the VOCs oxidation with the OH radical strengthening the process.
The Amazon is the largest and most diverse rainforest in the world, comprising about 390 billion broadleaf trees of 16,000 distinct species (Ter Steege et al., 2013). The Amazon basin encompasses about 7 million $km^2$, including territories of Brazil, Bolivia, Peru, Ecuador, Colombia, Venezuela, Guyana, Suriname, and French Guiana, with a significant portion almost untouched by

human activity with their natural environmental features preserved. The BVOCs mixing ratios in the Amazon are variable, with values ranging from 2.4 to 7.8 ppbv, depending on location, altitude, and seasonal behavior of radiation, temperature, and phenology (Yáñez-Serrano et al., 2015 and references therein). Harley et al. (2004), for example, estimated that about 38% of the plants in the Amazon forest emit isoprene. Also, studies have shown that the capacity of plants for producing and storing isoprenoids is very specific (Laothawornkitkul et al., 2009; Sharkey et al., 2008).

The atmosphere of the Amazon, in its undisturbed state, oxidizes the BVOCs naturally emitted by the forest vegetation, recycling some OH and depositing reactive carbon back to the surface as several oxidation products, including SOA. In this way, the cleaning process also acts as a local recycling mechanism, preventing the loss of essential nutrients from the forest (Lelieveld et al., 2008). It is estimated that about 90% of the isoprene and 50% of the terpenes (($C_5H_8$)$_n$) are removed from the atmosphere via oxidation by OH, followed by the deposition of oxidized VOC and SOA within a time scale of a few hours (Monks, 2005). In fact,

the isoprene is an important compound in the atmospheric chemistry over forest regions because of its abundance and high reactivity with OH (Barket et al., 2004; Prinn, 2014).





For several years, the traditional understanding was that the unpolluted atmosphere, defined by low levels of nitrogen oxides (NO$_x$), has low concentrations of OH during the midday, typically $1–5 \times 10^5$ molecules cm$^{-3}$; however, known discrepancies between atmospheric chemistry model results and observations raised the supposition of a missing OH source (Warneke et al., 2001; Whalley et al., 2013). Recently, airborne measurements performed in an unpolluted atmosphere over the Amazon rainforest found unexpected high oxidative capacity levels, which, complemented with laboratory and numerical modeling studies, led to a different hypothesis for OH production (Lelieveld et al., 2008). Concentrations of OH around 5.6 ($\pm$1.9) $\times 10^6$ molecules cm$^{-3}$ were measured in the planetary boundary layer (PBL) over the Amazon, concomitant with CO, NO, and O$_3$ mixing ratios of 113 ($\pm$13.9) ppbv, 0.02 ($\pm$0.02) ppbv, and 18.5 ($\pm$4.6) ppbv, respectively, values typical of the unpolluted atmosphere. This work pointed to the reaction of isoprene with peroxy radicals (HO$_2$) as an alternative pathway to OH production in an unpolluted environment (Lelieveld et al., 2008). Other OH observation studies conducted in pristine rainforests showing low-NO and high isoprene have consistently reported unaccountably high OH levels, e.g. (Whalley et al., 2011). Rohrer et al. (2014) compiled several previous OH observations in environments characterized by large VOC concentrations, such as forested areas, and concluded that it requires a substantial OH recycling mechanism to reconcile the discrepancy between observations and model outcomes based on the conventional understanding of isoprene photo-oxidation (Logan et al., 1981). However, a different school of thought considers these discrepancies between model and observation of OH production due to instrument artifacts. Mao et al. (2012) directly demonstrated the magnitude of potential instrument artifacts by adapting a novel background characterization method called a chemical removal technique, a method to measure OH in parallel with the traditional Fluorescence Assay with Gas Expansion (FAGE). The study also illustrated that the application of the chemical removal technique results in agreement between observed and model-calculated diurnal OH variations based on the conventional isoprene photo-oxidation. The same research group also deployed this instrumentation in a rural Alabama forest site as a part of the Southern Oxidant and Aerosol Study (SOAS) campaign (Feiner et al., 2016) and found high isoprene concentrations (up to 10 - 20 ppb) and low-NO levels (~ 50 ppt) in the afternoon. In this photochemical environment, the observed OH with the chemical removal technique agrees well with the model-calculated OH based on the conventional isoprene photo oxidation scheme. More recently, significant advances have been made with organic peroxy radicals (RO$_2$) produced as intermediates of atmospheric photochemistry, showing the importance of both RO$_2$ and NO pathways to isoprene chemistry, even in low-NO regions. An accurate understanding of the two pathways is required for quantitative predictions of particulate matter concentration, oxidation capacity, and consequent environmental and climate impacts (Liu et al., 2016).

Although Amazonia is mostly dominated by pristine areas, commonly described as low-NO region, there are regions that have been strongly impacted by human activity. The most devastating example is the ongoing deforestation, followed by vegetation burning to open areas for pasture and agriculture production. During the austral winter (from July to October), the Amazonia climate is typically dry and is disturbed each year by extensive vegetation fires in areas of deforestation and agricultural or pasture land management, particularly along the so-called deforestation arc, an area of about 500,000 km$^2$ extending from the southwestern to the eastern border of the forest (Artaxo et al., 2013). During the fire events, an intricate myriad of chemical and physical processes occurs. The continuous increase in temperature of the fresh biomass caused by nearby fires can distill species



absorbed by plants with low boiling point (*e.g.*, $T_{isoprene} \cong 307$ K), macromolecular bonds can be broken (i.e., low-temperature pyrolysis), gasification reactions converting carbon in the solid char to CO and $CO_2$ can occur and the flames efficiently oxidize the volatile gases to species such as $H_2O$, $CO_2$, and $NO_x$ (Bertschi et al., 2003; Longo et al., 2009). The release of isoprene and other BVOCs is dependent on the different phases of biomass combustion, and diverse vegetation communities affect the amount

and diversity of volatile organic compounds released (Ciccioli et al., 2014). In this disturbed atmosphere, the assumed natural efficient OH recycling mechanism is affected, altering the oxidative capacity of the atmosphere.

In the absence of biomass burning emissions, isoprene is the dominant reactive VOC in the pristine Amazon forest, and during the day, isoprene oxidation dominates the OH chemistry producing, among other products, methyl vinyl ketone (MVK), methacrolein (MACR), and hydroxyhydroperoxides (ISOPOOH) (Karl et al., 2007; Liu et al., 2016; Rivera-Rios et al., 2014). In a smoky

atmosphere, isoprene oxidation also produces mainly MVK and MACR, however, the molar yields can slightly differ from the ones for the unpolluted condition (i.e., low levels of NO). The updated chemistry of isoprene degradation in the Master Chemical Mechanism (MCM v3.3.1, Jenkin et al., 2015) reported molar yields of about 47% and 34% for MVK, and 20% and 23% for MACR, in low (0.1 ppbv) and high (10,000 ppbv) NO level environments, respectively.

In the context of an Amazon rainforest impacted by anthropic and by biogenic emission sources, the airborne measurements

conducted in Amazonia during the South American Biomass Burning Analysis (SAMBBA) in 2012 included several fire emissions tracers, as well as isoprene and its oxidation products. SAMBBA flights were carried out in both regions not directed and directed affected by fire emissions. In this work, we analyzed SAMBBA measurements to assess the impact of the smoke on the oxidative capacity of the atmosphere in the Amazon region. Due to the lack of direct measurements of OH during SAMBBA, we used the ratio of the mixing ratios of isoprene oxidation products (MVK, MACR and ISOPOOH) to isoprene as a proxy for the

OH levels.

Motivated by the discrepancies between model and observation of OH production in the atmosphere and the influence of the biomass burning plumes in the isoprene reactivity with OH during the day, we propose in this study a refinement in the OH estimation method that has been applied by several previous studies (Apel, 2002; Karl et al., 2007; Kuhn et al., 2007; Stroud et al., 2001).

The paper is structured as follows. In section 2, we present SAMBBA field campaign, including the meteorological conditions and fire occurrence during the campaign period, along with the airborne measurements discussed in this study. The classification method of flight tracks, as well as the method for the indirect OH calculation, are also covered in section 2. In section 3, we presented and discussed the ambient distribution of chemical compounds in the atmosphere (CO, $NO_x$, $O_3$, and isoprene) during SAMBBA, the factors that affected the ratio [MVK+MACR+ISOPOOH]/[isoprene] and the oxidative capacity in distinct

environments. Finally, in section 4 the main findings are summarized.



## 2 Observations and method of analysis

### 2.1 SAMBBA field campaign

SAMBBA field campaign was an airborne experiment carried out in the Brazilian Amazonian sector late in the dry season and
during the transition from the dry to the wet season, from the 14th September to the 3rd October 2012. Numerous atmospheric measurements were conducted onboard the BAe-146 research aircraft, during 20 research flights and 67 flight hours. The BAe-146 research aircraft, from the Facility for Airborne Atmospheric Measurements (FAAM - http://www.faam.ac.uk), was based in Porto Velho - RO, but made use of other regional airports (Palmas - TO, Rio Branco - AC, and Manaus - AM airports) to extend the operational range of the aircraft (Figure 1). During SAMBBA, the areas with positive anomalies of precipitation were mostly
in western and central Amazonia, while the eastern sector was drier than the climatic average. The mean daily rainfall east of SAMBBA flight area was typically below 1 mm. In contrast, in the western and central part, the mean daily precipitation ranged from 3 to 10 mm because of an intense cold front incursion, an early precursor of the dry-to-wet transition season. As a result, the fires in the western part of Amazonia, where most of SAMBBA flights took place, were scattered and intermittent. The most intense and persistent fire activity occurred in the eastern part. The aerosol optical depth in Porto Velho dropped from the typical
1.5 (channel 550 nm) in the first half of September 2012 to below 0.5 during SAMBBA; in contrast, the AOD was constantly above 1 in the eastern part of Amazonia.

The BAe-146 research aircraft flew with a comprehensive suite of instrumentation, measuring aerosols and cloud microphysics properties, chemical tracers, radiative fluxes, and several meteorological variables. Essential for this work were measurements of isoprene, MVK, MACR and ISOPOOH, which were carried out using an onboard proton transfer reaction mass spectrometer
(PTR-MS, Ionicon, Innsbruck, Austria) with a quadrupole detector and a typical cycle time around 3–5 s. The instrumental, operational and calibration details are described in Murphy et al. (2010), but it is pertinent to note that (1) the quadrupole detector cannot distinguish between the isobaric molecules MVK, MACR and ISOPOOH, so it reports the data at m/z 71 as the sum of 3 isomers, even though it was only calibrated for MVK+MACR (Liu et al., 2016), and (2) there is a well-known interference in the isoprene signal at m/z 69 in biomass burning plumes from furan. The PTR-MS was calibrated post-flight using a calibrated gas
standard provided by Apel-Reimer. We compared the PTR-MS isoprene data with the isoprene data derived from the whole air sampling (WAS) system on the aircraft to correct the PTR-MS isoprene data due to a probable interference from furan at m/z 69 in biomass burning plumes. The WAS system was described in Hopkins et al. (2011) and consists of discrete air samples collected in 3-liter silco-treated stainless steel canisters with subsequent post-flight analysis by GC-FID. In the background environment, the agreement between the two systems was excellent (Isoprene$_{was}$/Isoprene$_{ptrms}$ = 0.81, SD=0.56), while in biomass burning
regions we estimated a high furan contribution in fresh (Isoprene$_{was}$/Isoprene$_{ptrms}$ = 0.25, SD=0.12) and aged (Isoprene$_{was}$/Isoprene$_{ptrms}$ = 0.77, SD=0.57) smoke plumes. The isoprene data has been adjusted accordingly.

In addition, NO measurements were conducted using a chemiluminescence instrument (Air Quality Design Inc., Wheat Ridge, CO, USA), with the $NO_2$ measured using a second channel after photolytic conversion to NO. The photolytic conversion eliminates the possible interference from $NO_z$ on the $NO_2$ channel. The detection limits were close to 10 pptv for NO and 15 pptv



for $NO_2$ for 10 s averaged data, with estimated accuracies of 15% for NO at 0.1 ppbv and 20% for $NO_2$ at 0.1 ppbv (Allan et al., 2014). For the $O_3$ and CO analyses, we used the TEi49C and AL5002 VUV Fast Fluorescence onboard instruments, respectively (Gerbig et al., 1996, 1999; Palmer et al., 2013). Calibration gases were supplied to the rack from the gas bottle stowage, and the air sampling from the atmosphere was via the air sample pipes and a dedicated window-mounted inlet system.

## 2.2 Classification method of flight tracks

During the planning phase, SAMBBA flights were classified according to their scientific objectives as either biogenic or biomass burning flights (Figure 1). For this study, we selected 13 flights according to the gaseous chemistry data available (Table 1). Additionally, we only considered the data collected below 2,000 m and between 11:00 am and 6:00 pm to capture the difference in the oxidative capacity activity along the altitude during daytime, since the OH concentration is regulated by photochemistry

(Elshorbany et al., 2009). Despite the classification in the planning phase, parts of some flight tracks passed through unpolluted regions, smoke haze, or even interception of fresh smoke plumes. To maximize the use of data, we classified parts of the flight tracks according to the CO mixing ratio values as background (BG) and biomass burning. According to Andreae et al. (2012) and several references therein, the Amazon rainforest atmosphere has a background CO mixing ratio typically around 100 ppbv. However, the mean CO inflow into the Amazon Basin during SAMBBA period at 500 hPa, retrieved from Atmospheric Infrared

Sounder (AIRS) measurements onboard the AQUA satellite, ranged between 140 and 160 ppbv (Figure 2). This hemispheric inflow is homogeneous along the vertical column up to around 400 hPa. In fact, there were only few SAMBBA samples with CO mixing ratio values below 100 ppbv. Therefore, we adopted a threshold of 150 ppbv to represent the background of CO in the Amazon atmosphere during SAMBBA campaign.

As $O_3$ is formed photochemically downwind during smoke aging, the enhancement ratio of $O_3$ to CO is acceptable as a reliable

indicator of the smoke plume age (Andreae et al., 1994; Parrish et al., 1993). Furthermore, due to the lack of $NO_y/NO_x$ ratio in SAMBBA, we used the ratio of $O_3$ to CO as a proxy for smoke plume age. The biomass burning flights tracks with [CO] > 150 ppbv were then reclassified as fresh smoke plume (FP) or as aged smoke plume (AP) interceptions according to the following:

$$ER_{\Delta O_3/\Delta CO} = \frac{[O_3]_{smoke} - [O_3]_{background}}{[CO]_{smoke} - [CO]_{background}} \qquad (1)$$


During the BG flight tracks (CO ≤ 150 ppbv), the mean value of the $O_3$ mixing ratios near the surface (< 500 m) was 21 ±7 ppbv, which we then adopted as the $O_3$ mixing ratio background. In Table 2, we list the values of $ER_{[\Delta O_3]/[\Delta CO]}$ and the estimated smoke plume age for several smoke measurements in Amazonia and Africa. Jost et al. (2003) found the value of $ER_{[\Delta O_3]/[\Delta CO]} = 0.1$, two hours after emission in Otavi, northern Namibia; and Andreae et al. (1988) found 0.08 for fresh biomass burning (650 m altitude) 195 in the Amazon Basin region. Comparable values of $ER_{[\Delta O_3]/[\Delta CO]}$ (0.09) were observed in other young smoke plumes with 0.5–1.0



hour aged during the Southern African Regional Science Initiative 2000 - SAFARI 2000 (Hobbs et al., 2003; Yokelson et al., 2003). Mauzerall et al. (1998) reported 0.15 for fresh smoke plumes with fewer than 4.8 hours over regions with active fires in the Northeast region of Brazil and in Africa. In short, we classified the parts of the flight tracks as background (BG) when [CO] ≤ 150 ppbv, and the biomass burning flights ([CO] > 150 ppbv) into two subgroups of fresh smoke plumes (FP), with [O$_3$]/[CO] < 0.1,

and aged smoke plume (AP), with [O$_3$]/[CO] ≥ 0.1.

### 2.3 Method description for the OH calculation

The OH concentrations from the [MVK+MACR+ISOPOHH]/[Isoprene] ratio using the sequential reaction model were originally developed by Apel, 2002 and Stroud et al., 2001 and modified according to the approach of Karl et al. (2007). This method can be

used to investigate the impact of vertical transport, representing the processing time of the isoprene and its oxidation products from the surface to the atmosphere through the ratio of PBL depth and the convective velocity scale. To have a more accurate OH estimation, we modified the processing time $t$ to represent not only the vertical transport but also the horizontal atmospheric circulation, where $t$ was calculated as a function of the enhancement ratio $ER_{[\Delta O_3]/[\Delta CO]}$. Table 2 shows the plume age time and the enhancement ratio $ER_{[\Delta O_3]/[\Delta CO]}$ values used in the new approach. This method is based on observations that the isoprene reaction

rate with OH (rate coefficient $\cong 1.0 \times 10^{-10}$, lifetime $\cong 1.4$ h) is more important than with O$_3$ (rate coefficient $\cong 1.3 \times 10^{-17}$, lifetime $\cong 1.3$ d) during the daytime. Following the simplified sequential reaction model, we can estimate OH concentration, in molecules cm$^{-3}$, with the following analytical expression:

$$[OH] = \left[ \ln\left( 1 + \left( \frac{\frac{[MVK+MACR+ISOPOOH]}{[Isoprene]} * (K_{iso} - K_{prod})}{K_{iso} * 0.55} \right) \right) \right] / ((K_{iso} - K_{prod}) * t) \tag{2}$$


where k$_{iso}$ and k$_{prod}$ are, respectively, the reaction rate constants of isoprene + OH ($1.1 \times 10^{-10}$ cm$^3$ molecules$^{-1}$ sec$^{-1}$), and [MVK+MACR+ISOPOOH] + OH ($6.1 \times 10^{-11}$ cm$^3$ molecules$^{-1}$ sec$^{-1}$), and we are assuming a total yield of 0.55 of MVK+MACR+ISOPOOH from the OH + Isoprene reaction (Apel, 2002; Karl et al., 2007). We estimated the processing time as $t = 5.3e^{5.4 * ER_{[\Delta O_3]/[\Delta CO]}}$ (seconds), which is the fitting function of several previous measurements of $ER_{[\Delta O_3]/[\Delta CO]}$ and plume age

observations in tropical and subtropical sites (Table 2).



## 3 Results and discussion

### 3.1 Ambient distributions of CO, NOₓ and O₃

Figure 3 depicts CO, $NO_x$, and $O_3$ mixing ratios measured at different altitudes, up to 2 km, and time of the day, between 11:00

am and 6:00 pm. In Figure 3, the flight tracks are separated according to the BG, FP, and AP classification, while Table 3 shows typical values of CO, $NO_x$, and $O_3$ mixing ratios measured in this study and during several previous airborne campaigns in Amazonia and savannah areas in Brazil. During SAMBBA field experiment, in BG conditions (i.e., CO < 150 ppbv), the $NO_x$ mixing ratio ranged from 50 pptv to 200 pptv. Torres and Buchan (1988) reported measurements of NO mixing ratios ranging between 20 and 35 pptv during the Amazon Boundary Layer Experiment (ABLE-2A) between July and August 1985. Modeling

results of Jacob and Wofsy (1988) found $NO_x$ mixing ratio values around 200 pptv, with the NO mixing ratio values similar to the ABLE-2A observations that were conducted over the Amazon rainforest. Also in the Brazilian Amazon Basin during the wet season, aircraft measurements as part of the NASA Atmospheric Boundary Layer Experiment (ABLE 2B), showed $NO_x$ mixing ratios ranging from 4 – 68 pptv (Singh, 1990). Comparing our results with these previous studies, SAMBBA experiment showed a slight influence from polluted regions. More recently, Liu et al. (2016) used four sets of different Master Chemical Mechanisms

and estimated that NO mixing ratio using the NO vs $HO_2$ isoprene chemistry ($fHO_2$:$fNO \sim 0.6 – 1.4$) would be around 20 – 40 pptv of NO based on measurements in the Amazon, lower than that obtained in our study. On the other hand, flight tracks in biomass burning areas showed high values of $NO_x$ in FP (50-1,250 pptv) and AP (50-950 pptv) compared with other studies in forest areas of Amazonia, including a study in the cerrado area in Brazil (~ 750 pptv) conducted by Crutzen et al. (1985) during the dry season.

The $O_3$ mixing ratios in the BG environment reached 40 ppbv at about 600 m altitude during flight B735 (at 11:30 am, Figure 3), although typical values ranged from 10 to 45 ppbv (< 2000 m) (Table 3). This value of 40 ppbv at 600 m altitude is nearly two times the mean value of the $O_3$ mixing ratio that we used as background in the enhancement ratio, and even out of the range of the standard deviation (21 ppbv, SD=7). As a secondary pollutant, $O_3$ is commonly found in low concentrations near the surface, a fact not observed in our study even for the BG samples. Bela et al. (2014) reported $O_3$ ranging from 10 to 20 ppbv during the wet-

to-dry transition season (May/June 2009), measurements performed in a clean atmosphere during the Regional Carbon Balance in Amazonia (BARCA-B) campaign. In contrast, during the BARCA-A campaign in November/December 2008 (during the dry-to-wet transition season), the $O_3$ mixing ratio ranged from 40 to 60 ppbv in an area influenced by fires, values which are comparable with FP (10-75 ppbv) and AP (20-70 ppbv), and even in BG conditions (10-45 ppbv) during SAMBBA.

In terms of CO and $NO_x$, flight tracks classified as FP were the most polluted of the campaign. The CO and $NO_x$ mixing ratios for

FP reached, respectively, values above 3,000 and 60 ppbv at 600 m from the surface between 11:00 am and 12:00 pm. The enhancement of CO and $NO_x$ mixing ratios near the surface suggests significant vertical transport due to the hot plume buoyancy, which may increase the tracer lifetime released to the atmosphere. The vertical transport can be observed mainly for CO at different altitudes and time of day, since the CO is preserved longer along the plume when compared with $NO_x$. The measurements in fresh biomass burning plumes also captured higher levels of CO mixing ratios (≅500 ppbv) at 1.4 km and 2 km




of altitude. Yokelson et al. (2007), during the Tropical Forest and Fire Emissions Experiment (TROFFEE), reported a vertical transport mechanism called a "mega-plume" at 2 km altitude during a flight south of the Amazon rainforest (from Manaus to Cuiabá), with the CO mixing ratio reaching 1,200 ppbv. The TROFFEE experiment used airborne measurements during the 2004 Amazon dry season, and reported, on 8 September, the presence of a massive plume formed by numerous fires. During SAMBBA campaign, we also detected the presence of similar plumes during the flight B742, classified mostly as FP, with a unique CO

mixing ratio value, peaking ~5,000 ppbv at about 600 m. These results demonstrate the strength of vertical transport during a fresh biomass burning event, with the plume injection height up to 2 km. Freitas et al. (2006, 2007, 2010) highlighted the importance of representing the injection height of biomass burning plumes in numerical models to describe the regional smoke distribution. Trentmann and Andreae (2003) also demonstrated a large impact of fire emissions on the chemical composition in a young biomass burning plume using a 3-D chemical transport model and direct observations. These authors reported simulated high

values near the fire ($z \cong 150$ m) for CO and $NO_x$, with mean values around 18,000 ppbv and 404 ppbv, respectively. On the other hand, Andreae et al. (2012) reported CO mixing ratios up to 400 ppbv in a smoky region in the southern Amazon Basin during the BARCA-A experiment. The highest values were found at about 1,000 m altitude, late in the dry season (November 2008). In this study, the CO mixing ratio measurements in FP and AP ranged from 150 to 900 ppbv and from 150 to 450 ppbv, respectively. The CO mixing ratios in FP (150-900 ppbv) are comparable with values found by Reid et al. (1998) in a cerrado area (440-763 ppbv)

and some forest studies (up to 600 ppbv) conducted by Andreae et al. (1988), Kaufman et al. (1992) and Yokelson et al. (2007). The values found in AP agreed with values of CO mixing ratios from forest areas, impacted by smoke haze plumes (Table 3).

In Figure 3, during most of the flight tracks classified as AP, the $NO_x$ mixing ratio values were below 2 ppbv, except for a peak of 6 ppbv at 600 m from the surface between 11:30 am and 12:00 pm, a value below that observed in the FP environment (60 ppbv). Conversely, the $O_3$ mixing ratio results in FP presented high levels ($\cong$ 80 ppbv) around 12:10 pm at 1,300 m altitude. We also

investigated the high levels of CO and $NO_x$ mixing ratios found in FP corresponding to the same flight in which we also detected high levels of $O_3$ in AP. In fact, Figure 4 shows the track of flight B742 in transition from a remote site impacted by FP to an urban site impacted by AP in Palmas-TO ($O_3 \cong$ 80 ppbv). The mean value of $O_3$ found in FP was 31 ppbv (SD = 14), which is 29% lower than measured for AP (44 ppbv, SD = 13 ppbv), since $O_3$ is produced as a secondary product from the interaction between VOCs and $NO_x$. The $O_3$ mixing ratios in FP peak at about 60 ppbv near the surface (200–600 m) and, for most cases,

plumes with about 40 ppbv were observed both near the surface and in high altitudes (Figure 3). During TROFFEE experiment, Yokelson et al. (2007) found $O_3$ mixing ratios of about 30 ppbv in smoke haze layers in Amazonia. Our results showed higher values for $O_3$ mixing ratios, ranging from 10 to 75 ppbv in FP and 20 to 70 ppbv in AP. Agreeing with SAMBBA results, Reid et al. (1998) found $O_3$ mixing ratios ranging from 60 to 100 ppbv during SCAR-B experiment in the 1995 dry season, and Kaufman et al. (1992) found similar levels of $O_3$ in a forest site in Amazonia during BASE A (Table 3).






### 3.2 Isoprene and its oxidation ratio

Information about the isoprene transport and chemistry can be derived from the isoprene abundance in the atmosphere and the ratio of its oxidation products over isoprene, [MVK+MACR+ISOPOOH]/[isoprene]. During the day, the isoprene chemistry is affected mainly by the distance from the emission source (transport time), photochemical degradation and availability of OH,

which react with isoprene to produce (among other chemical species) MVK, MACR and ISOPOOH (Kuhn et al., 2007; Liu et al., 2016). During SAMBBA, the mean isoprene mixing ratio in BG was 2.8 ppbv and 1.5 ppbv for the boundary layer and cloud layer, respectively. We also detected higher values of $O_3$ (40 ppbv at 600 m) in BG, shown in Figure 3, coinciding with the interpolated cross-section of isoprene ($\leq$4 ppbv at 600 m), also in the BG environment (Figure 5). As mentioned by Barket et al. (2004), a sequence of reactions initialized by the reaction of isoprene with OH leads to the production of organic peroxy radicals

($RO_2$), which then react with $NO_x$ promoting the $O_3$ formation observed during the BG flights tracks. Table 4 summarizes the mean values of isoprene and the oxidation ratio [MVK+MACR+ISOPOOH]/[isoprene] during SAMBBA, and previously reported airborne measurements in remote areas and biomass burning environments. The isoprene mixing ratios measured during SAMBBA in the BG environment agree with values reported in pristine areas of the Amazon forest (Greenberg et al., 2004; Greenberg and Zimmerman, 1984; Gregory et al., 1986; Helmig et al., 1998; Kuhn et al., 2007; Lelieveld et al., 2008; Rasmussen

and Khalil, 1988; Zimmerman et al., 1988). Some studies conducted in the Amazonian tropical forest (e.g., Greenberg et al. 2004 and Kuhn et al. 2007) reported isoprene mixing ratio up to ~ 7 ppbv, values higher than we found during SAMBBA campaign.

On average, we found a reduction of isoprene in FP (1.4 ppbv), and a more discrete reduction in AP (2.4 ppbv), relative to the BG value (2.8 ppbv), within the PBL (<1,200 m), producing a value around 50% and 14%, respectively. In contrast, we observed above the PBL (> 1,2000 m) an impressive increase of about 60% of the isoprene in AP (2.4 ppbv) relative to the BG value (1.5

ppbv), which is about the same mean value found in FP (1.6 ppbv). These high levels of isoprene at higher altitudes in air masses affected by biomass burning emissions are likely to be associated with the heat released from vegetation fires affecting nearby plants with enough energy to release significant amounts of isoprene to the atmosphere, especially in tropical forest fires in Brazil (Ciccioli et al., 2014). Müller et al. (2016), for example, found isoprene mixing ratios up to 15 ppbv in a smoke plume from a small forest fire in Georgia, USA. We also found higher isoprene mixing ratios in the upper levels (> 1,200 m) of smoke areas

when compared with pristine mixed layer studies mentioned previously. These results reinforce the hypothesis that fire activity promoted the isoprene transport to higher altitudes both in fresh ($\cong$ 6 ppbv, 1,700–2,000 m) and aged plumes ($\cong$ 4 ppbv, 1,600– 2,000 m) during SAMBBA flights (Figure 5). In Figure 6, we also verified the average isoprene mixing ratio (< 2,000 m) in AP (2.4 ppbv) was 71% higher than FP (1.4 ppbv) and similar to the mean value measured in BG (2.6 ppbv).

In fresh smoke plumes, biomass burning tracers, such as acetonitrile and acetaldehyde, are present at high concentration, while the

[MVK+MACR+ ISOPOOH]/[isoprene] ratio is low. In contrast, aged smoke plumes typically have higher values for the [MVK+MACR+ISOPOOH]/[isoprene] ratio, since there is more time for the isoprene degradation. Figure 7 presents the plume interception during the flight B732, between 10:00 am and 11:30 am, in which it is possible to observe the difference between FP and AP interceptions through the biomass burning tracers and [MVK+MACR+ISOPOOH]/[isoprene] ratio. In this study, we



found the [MVK+MACR+ISOPOOH]/[isoprene] ratio mean value ranging from around 1.7 in the boundary layer up to 3.3 in the

cloud layer for BG conditions, with AP presenting a similar value (2.3) for both boundary layer and cloud layer. In contrast, FP had the highest value in the boundary layer (7.0) and cloud layer (6.1), values reported by Kuhn et al. (2007), in the tropical forest in Brazil (north of Manaus).

We did not find any substantial variation above the PBL in [MVK+MACR+ISOPOOH]/[isoprene] ratio associated with the presence of smoke in AP (2.8), but did find an increase in the BG value (3.3) in the upper levels (> 1,200 m). The FP is more

active within the boundary layer than in upper levels, with the isoprene oxidation ratio about 6.1 above 1,200 m. Comparable with our results in FP, Kuhn et al. (2007) during the Cooperative LBA Airborne Regional Experiment (LBA-CLAIRE-2001), also found [MVK+MACR+ISOPOOH]/[isoprene] ratio values up to 2, below 1,000 m of altitude, and between 2 and 10, within the 1,000–2,000 m vertical layer. In summary, we found a strong increase of the isoprene oxidation ratio from the surface up to 2,000 m for FP relative to the BG and AP observed during SAMBBA and other previous studies in biomass burning environments

(Table 4). The energetic process that occurs during the biomass burning causes the isoprene plume to be transported rapidly to higher levels, impacting the isoprene oxidation level, with FP samples presenting a higher [MVK+MACR+ISOPOOH]/[isoprene] ratio.

We also observed during SAMBBA campaign values of the [MVK+MACR+ISOPOOH]/[isoprene] ratio above 6 in BG air masses above 1,800 m and between 12:00 pm and 13:30 pm. In contrast, in FP and AP, values in the range 4-6 are equally

distributed in the vertical profile, with some high values near the surface (Figure 5). Along the cloud layer (1,200–2,000 m), we found that isoprene oxidation in BG environment increase (94%) as in AP levels (Table 4). Karl et al. (2007) also reported evidence of an increase of the oxidizing power of the atmosphere in the transition from PBL to cloud layer (1,200–1,900 m) during TROFFEE experiment, with the [MVK+MACR+ISOPOOH]/[isoprene] ratio ranging from 0.39 up to 1.2 between 300 m and 1,800 m, already into the cloud layer (CL). Although lower values for the [MVK+MACR+ISOPOOH]/[isoprene] ratio were

found during TROFFEE compared with LBA-CLAIRE, both studies have suggested the occurrence of an oxidizing power in the transition     from     the     PBL     to     the     CL.     In     both     cases,     there     was     a     positive     gradient,     increasing     the [MVK+MACR+ISOPOOH]/[isoprene] ratio. Furthermore, Helmig et al. (1998) reported similar behavior in a remote Peruvian Amazonia site, with the [MVK]/[isoprene] ratio equal to 0.15, 0.19, and 0.48, near the surface ($\cong$ 2 m), in the PBL (91–1,167 m), and above the PBL (1,481–1,554 m), respectively. The [MVK+MACR+ISOPOOH]/[isoprene] ratio increasing toward the top of

the PBL and CL is likely to be due to the enhancement of the photolysis rates. Direct experimental data reported by Mauldin et al. (1997) also indicated significant changes above and inside cloud decks due to cloud edge effects on photolysis rates that have a major impact on OH production rates. Figure 8 presents the density distribution for the [MVK+MACR+ISOPOOH]/[isoprene] ratio along several altitude layers during SAMBBA. In FP, the average [MVK+MACR+ISOPOOH]/[isoprene] ratio was 6.3 at 500–1,000 m, 7.6 at 1,000–1,500 m, and returning to 5.9 at 1,500–2,000 m. These results are consistent with the increase of the

oxidative capacity in the transition from the PBL to the CL, reported by Mauldin et al. (1997) and Karl et al. (2007), with SAMBBA measurements showing an average [MVK+MACR+ISOPOOH]/[isoprene] ratio constantly increasing in BG and the highest value at 500 – 1,000 m in AP. The results showed the isoprene oxidation reaction is also enhanced at higher altitudes in



the BG environment, increasing from 1.4 at the first 500 m to 3.4 at 2,000 m. Another characteristic observed during biomass burning events is their capacity to disturb the isoprene oxidation reactions, especially in the fresh plumes. As showed in Figure 8, the isoprene oxidation is higher in fresh smoke plumes at lower altitudes (~ 500) than in aged smoke, anticipating near the surface a complex chain of oxidation reactions which may be related to SOA formation. Rohrer et al. (2014) compared observations of OH radicals in different environments characterized by high VOC concentrations; they found that VOC degradation not only accelerates but also occurs at the maximum rate if $NO_x$ is present in adequate amounts. Thus, the biomass burning is a source of $NO_x$, favoring the increase of the oxidative capacity. According to Rohrer et al. (2014), the OH recycling mechanism is shown to be active not only in pristine biogenic air masses but also in the interface region between anthropogenic and biogenic emissions, such as in the region surrounding Manaus - AM, where urban and biogenic emissions are mixed.

### 3.3 OH predicted using sequential reaction approach

The abundance of OH in the atmosphere is determined by equating the kinetic rate of its production and loss. Due to the absence of OH measurements during SAMBBA, we inferred the OH concentrations using a sequential reaction model to the observed profiles of the [MVK+MACR+ISOPOOH]/[Isoprene] ratio. Table 4 presents the average values of OH concentration in remote areas and biomass burning environments worldwide, and the estimated OH concentration calculated via the sequential reaction model approaches from Karl et al. (2007) and via the approach proposed in this study. In FP, the estimated OH concentration reached the highest value within the PBL ($1.4 \times 10^6$ molecules $cm^{-3}$), when compared with AP or even with the BG environment, both with OH concentration $\sim 0.1 \times 10^6$ molecules $cm^{-3}$. The photochemical environment in young biomass burning plumes differs from the clean conditions, especially near the surface. Hobbs et al. (2003) found OH concentrations of about $1 \times 10^7$ molecules $cm^{-3}$ for a fresh plume from savanna fire in South Africa, value higher than those found in our estimation. In CL, the estimated OH in AP showed a reduction of 15% relative to BG ($0.5 \times 10^6$ molecules $cm^{-3}$) environment, in opposition to increased pattern in FP ($1.2 \times 10^6$ molecules $cm^{-3}$). The estimated OH concentration corroborates the hypothesis that the biomass burning event can intensify the oxidative capacity at low altitudes.

Figure 9 shows the vertical profile of estimated OH concentration in different chemical regimes, comparing the sequential reaction model according to the original approach of Karl et al. (2007) with the new approach used in this work. Throughout altitude range 0 – 2,000 m, the difference in OH calculation between the two methods was approximately 2 orders of magnitude, although presented a similar pattern in the different chemical regimes. Flight tracks classified as BG tends to increase the OH concentration along the altitude, with the inflection point occurring before the 1,000 m. Differing from BG environments, FP present a decrease pattern for OH concentration after 1,000 m of altitude, with the AP in an intermediate state. Our results suggest that in the fresh plumes, the vertical transport predominates with the oxidative capacity reaching its maximum at 1,000 m. In the flight tracks classified as BG, we observed the widest variation in the average OH concentration using the new sequential reaction model (Figure 9, on bottom), especially in upper levels ($0.5 – 1 \times 10^6$ molecules $cm^{-3}$), although reported a lower confidence level in this





region due to a reduce number of samples. In all three different chemical regimes, the vertical profile of OH concentration presented an increase near to CL (~1,000 m), in agreement with previous studies (Karl et al., 2007; Kuhn et al., 2007; Langford et al., 2005; Mauldin et al., 1997).

The estimated OH concentration values presented in this study agree in order of magnitude with most modeled and observed values previously reported for Amazonia and other forest areas. Prediction studies in a forest site at Surinam, conducted by

Warneke et al. (2001), estimated a concentration of OH ranging from 1 to $3 \times 10^5$ molecules cm$^{-3}$ (24 h average), and Williams et al. (2001) calculated a range of 0.7-1.1 $\times 10^6$ molecules cm$^{-3}$ during daytime. During the Guyanas Atmosphere-Biosphere exchange and Radicals Intensive Experiment with a Learjet (GABRIEL) experiment in October 2005, the observed average OH concentration in the boundary layer (<1 km) over the Suriname rainforest in the afternoon was $4.4 \times 10^6$ molecules cm$^{-3}$ (Kubistin et al., 2010). On the other hand, Dreyfus et al. (2002) reported high levels of OH concentration (8-13 $\times 10^6$ molecules cm$^{-3}$) in the

boundary layer over a forest area in Sierra Nevada, California; this forest site was influenced by wind flow patterns, transporting anthropogenic volatile organic compounds and NO$_x$ from Sacramento region toward the Sierra Nevada.

Several studies investigated the uncertainties in the isoprene oxidation mechanism, and most of them focus on OH concentration levels through observational and modeling studies (de Gouw et al., 2006; Kubistin et al., 2010; Kuhn et al., 2007; Lelieveld et al., 2008; Lu et al., 2012; Whalley et al., 2014; Yokelson et al., 2007). Under a high isoprene and low-NO atmospheric regime, there

is a controversial discussion about the impact on the oxidative capacity in forest sites. Some observations indicate that high OH levels cannot be accounted for by the conventional OH production and recycling mechanisms (Rohrer et al., 2014), but some suggested that the enhanced OH signal is caused by instrumental artifacts rather than the ambient OH (Mao et al., 2012). Our calculated OH with the observational constraints is consistent with previous empirical estimates by Warneke et al. (2001) that estimated OH concentration around 1-3 $\times 10^5$ molecules cm$^{-3}$ (24 h average); Williams et al. (2001) also found 0.6-1.1 $\times 10^6$

molecules cm$^{-3}$ during daytime without augmented OH recycling mechanisms developed to account for the recent higher than expected OH observations. According to the most recent results in the Amazon rainforest (Liu et al., 2016), the order of magnitude of the OH concentration estimated in our study agrees well, with both OH concentrations close to $1 \times 10^6$ molecules cm$^{-3}$.

In this study, we observed an improvement in the estimated OH concentration values using the modified sequential reaction model

described in the section 2.3, for both biomass burning regimes and background environment. However, uncertainties exist associated with the lack of accuracy in dynamic factors in the simplified analytical expression (Eq. 2), such as vertical and horizontal transport, convective velocity above different vegetation cover, as well as the radiation regime influenced by clouds at high altitudes which are likely to affect the OH concentrations. We also evaluated the predominance of SAMBBA data between 11:00 am–2:00 pm, especially for BG and FP groups, may alter the distribution of [MVK+MACR+ISOPOOH]/[isoprene] ratio

along the diurnal cycle, and consequently modify the OH estimated. The change of the molar yield of the primary first-generation products of the OH-isoprene oxidation, as a function of NO mixing ratio, is another possible source of uncertainty in the estimation of OH concentrations.




## 4 Final remarks

We present a concise chemical characterization of the atmosphere of Brazilian Amazonia during SAMBBA airborne experiment
from 14th September to 3rd October 2012, comprising the transition period from the dry to wet season. SAMBBA flights were carried out in remote areas, as well as areas under the influence of biomass burning that commonly occurs in the region. The flight classification method adopted in this study prioritized the chemical regimes, using CO mixing ratios and the enhancement ratio of $O_3$ to CO to categorize different flight tracks to include BG flights, and FP and AP flights.

Measurements of CO, $NO_x$, and $O_3$ performed in areas not directly affected by local fire emissions reveals the signature of
biomass burning in the chemical composition of the background of the Amazonian atmosphere, due to long-range transport of biomass burning tracers both from Africa and the eastern part of Amazonia. In our analysis, we highlight the importance of photochemical age in areas influenced by biomass burning emission, with distinct results for FP and AP. Fresh smoke plumes had the highest mixing ratios of CO and $NO_x$, highlighting the strength of vertical transport through the detection of biomass burning products in the upper levels (> 1,200 m).

Regarding the isoprene, the measurements in the BG environment agree with values reported by several studies in pristine areas of the Amazon forest. Near fresh and aged smoke, we found much higher levels of isoprene both in fresh (6 ppbv, 1,700–2,000 m) and aged (4 ppbv, 1,600–2,000 m) smoke plumes. These results reinforce the hypothesis that fire activity has energy enough to promote the isoprene transport to higher altitudes, altering the isoprene oxidation mechanism when compared with remote areas. Fresh plumes also presented a higher [MVK+MACR+ISOPOOH]/[isoprene] ratio (7.0), when compared with both AP (2.3) and
BG (1.7), indicating a strong oxidation process within the boundary layer. Using the complementary approach of the simplified sequential reaction model used by Karl et al. (2007), we indirectly calculated the OH concentration modifying the processing time to represent not only the vertical transport but also the horizontal atmospheric transport time. This adjustment of the processing time provided reasonable OH concentration results close to those obtained in the recent GoAmazon campaign ($1 \times 10^6$ molecules $cm^{-3}$).

The highest value for OH in FP within the PBL ($1.4 \times 10^6$ molecules $cm^{-3}$) corroborates the results from [MVK+MACR+ISOPOOH]/[isoprene] ratio, confirming that the photochemical environment in young biomass burning plumes differs from the average conditions. We also detected a strong signal in the oxidative capacity at higher levels (~1,000 m), characteristic of the cloud layer existence, as reported by other studies (Karl et al., 2007; Mauldin et al., 1997).

For future research, we recommend further investigation of the impact of the dynamic factors in the estimation of OH mixing
ratios, such as horizontal transport and convective velocity above different vegetation cover, as well as the effect of the radiation regime influenced by clouds at high altitudes altering photolysis rates. Considering the recent updates in the molar yield change of the primary first-generation products of the OH-isoprene oxidation, we also expect a reduction in the uncertainties associated with the estimation of OH mixing ratio.



**Data availability**

SAMBBA field experiment data are available at Centre for Environmental Data Analysis (http://browse.ceda.ac.uk/browse/badc/sambba/data/faam-bae146) and complementary data are available on request.

**Author contributions**

Fernando C. dos Santos, Karla M. Longo and Alex B. Guenther prepared the manuscript. Fernando C. dos Santos analyzed the chemistry data and performed the estimation of OH density with contributions from Karla M. Longo, Alex B. Guenther and
Saewung Kim. Dasa Gu provided the WRF-Chem data used to calculate the estimated OH density. James R. Hopkins provided isoprene data from the whole air sampling (WAS) system canisters. James Lee provided NOx data. Dave E. Oram and Grant Forster provided data from Isoprene, Methyl Vinyl Ketone, Methacrolein and hydroxyhydroperoxide. Saewung Kim, Dasa Gu, Dave E. Oram, Grant Forster, James Lee, James R. Hopkins, Joel Brito, and Saulo R. Freitas reviewed the manuscript.

**Acknowledgements**

The Facility for Airborne Atmospheric Measurement (FAAM) BAe-146 Atmospheric Research Aircraft is jointly funded by the Met Office and Natural Environment Research Council and operated by DirectFlight Ltd. We would like to thank the dedicated efforts of FAAM, DirectFlight, INPE, the University of São Paulo, and the Brazilian Ministry of Science and Technology in making the SAMBBA measurement campaign possible. We thank Ben Johnson (Met Office) for his role in coordinating the SAMBBA campaign. Isoprene data from WAS sample analysis were provided by James R. Hopkins (National Centre for
Atmospheric Science and University of York). The São Paulo Research Foundation (FAPESP) supported this work through the projects 2012/13575-9, DR 2012/11676-2 and BEPE 2013/03391-0.

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



Table 1. SAMBBA research flights analyzed in this work. Reference locations indicated in the map of the Figure 2.

| Flight | Date | Take-off and Landing Hour (local time) | Region *Directions from Porto Velho - RO | Objectives |
|--------|------|----------------------------------------|-------------------------------------------|------------|
| B731 | 14 Sep 2012 | 10:00 14:35 | East | Biomass burning |
| B732 | 15 Sep 2012 | 10:30 14:40 | Surrounding Porto Velho-RO | Biomass burning |
| B734 | 18 Sep 2012 | 08:00 10:15 | Southeast | Biomass burning |
| B735 | 19 Sep 2012 | 08:00 11:40 | Northeast | Biogenic emissions |
| B737 | 20 Sep 2012 | 10:45 14:45 | Southeast | Biomass burning |
| B740 | 25 Sep 2012 | 7:45 11:00 | Surrounding Porto Velho-RO | Biomass burning |
| B742 | 27 Sep 2012 | 9:00 12:30 | Southeast Palmas-TO | Biomass burning |
| B744 | 28 Sep 2012 | 9:00 12:30 | Southeast | Biogenic emissions |
| B745 | 28 Sep 2012 | 14:00 17:30 | Southeast | Biogenic emissions |
| B746 | 29 Sep 2012 | 09:00 13:00 | East | Biomass burning |
| B748 | 02 Oct 2012 | 09:00 13:00 | East | Biomass burning |
| B749 | 03 Oct 2012 | 10:00 13:30 | Northwest | Biogenic emissions |
| B750 | 03 Oct 2012 | 15:00 18:30 | Northwest | Biogenic emissions |






Table 2. Observations of the enhancement ratio $ER_{[\Delta O_3]/[\Delta CO]}$ and plume age in tropical and subtropical sites.

| Tropics and subtropics region | Plume age | $^*ER_{[\Delta O_3]/[\Delta CO]}$ | Reference |
|---|---|---|---|
| Southern Africa | < 30 min | 0.09 | Hobbs et al., 2003 |
| Southern Africa | < 1 h | 0.09 | Yokelson et al., 2003 |
| Mexico | < 2 h | 0.08 | Yokelson et al. 2009 |
| Southern Africa | $\approx$ 2 h | 0.10 | Jost et al., 2003 |
| Brazil/Southern Africa | < 0.5 day | 0.15 | Mauzerall et al., 1998 |
| Brazil/Southern Africa | 0.5-1 day | 0.32 | Mauzerall et al., 1998 |
| Southern Africa | < 1 day | 0.01 | Yokelson et al., 2003 |
| Northern Africa | $\leq$ 2 days | 0.23 | Jonquières et al., 1998 |
| Southeast Asia | 2 – 3 days | 0.20 | Kondo et al., 2004 |
| Brazil/Southern Africa | 1-5 days | 0.71 | Mauzerall et al., 1998 |
| Southeast Asia | 4-5 days | 0.33 | Bertschi et al. 2004 |
| Brazil/Southern Africa | 5-7 days | 0.74 | Mauzerall et al., 1998 |
| South Africa/South America | $\leq$ 10 days | 0.75 | Singh et al., 2000 |
| Africa/South America | 10 days | 0.41 | Andreae et al., 1994 |

[*]The single value for enhancement ratio $ER_{[\Delta O_3]/[\Delta CO]}$ presented here represents the mean measurement.






Table 3. Airborne measurements of CO, $NO_x$, and $O_3$ mixing ratios in Amazonia and cerrado areas in Brazil.

| Month/Year | CO (ppbv) | NO$_x$ (pptv) | O$_3$ (ppbv) | Biome and reference |
|---|---|---|---|---|
| Sep/2012 | 135-150[a] | 50-200[a] | 10-45[a] | Forest and grassland, BG, this work |
| Sep/2012 | 150-900[b] | 50-1,250[b] | 10-75[b] | Forest and grassland, FP, this work |
| Sep/2012 | 150-450[c] | 50-950[c] | 20-70[c] | Forest and grassland, AP, this work |
| Aug/1979 | 70-500 | ~750 | 40-65 | Cerrado, Crutzen et al., 1985 |
| Aug/1980 | 100-400 | - | 20-55 | Forest, Crutzen et al., 1985 |
| Jul/1985 | 150-600 | 74-102 | 20-50 | Forest, Andreae et al., 1988 |
| Apr/1987 | 84-118 | 4-68 | 10-57 | Brazilian Amazon Basin, Singh, 1990 |
| Sep/1989 | 150-600 | - | 25-80 | Forest, Kaufman et al., 1992 |
| Sep/1992 | 100-400 | - | - | Forest, Blake et al., 1996 |
| Aug/1995 | 440-763 | - | 95-102 | Cerrado, Reid et al., 1998 |
| Aug/1995 | 482-566 | - | 61-70 | Forest, Reid et al., 1998 |
| Aug/2004 | 100-600 | - | 10-30[a, c] | Forest, Yokelson et al., 2007 |
| Nov/2008 | 100-300[b, c] | - | 40-60[c] | Forest, Andreae et al. (2012), Bela et al., 2014 |
| May/2009 | 60-110[a] | - | 10-20[a] | Forest, Andreae et al. (2012), Bela et al., 2014 |
| Mar/2014 | 100-150[d] | - | 10-60[d] | Tropical Forest, Brazil (west of Manaus), Liu et al., 2016 |

*Measurements in [a]background, [b]fresh plumes, and [c]aged plumes. Measurements with a background and maximum interval[d].*

*The data without an index was collected without any particular criteria.*





Table 4. Airborne measurements of isoprene, oxidation ratio [MVK+MACR+ISOPOOH]/[isoprene], and OH in remote areas and biomass burning environments worldwide.

| Month/Year | Isoprene (ppbv) | [MVK+MACR+ ISOPOOH]/ [Isoprene] | OH ($10^6$molec. cm$^{-3}$) | Biome, location, and reference |
|---|---|---|---|---|
| Sep/2012 | 2.8[c] 1.5[e] | 1.7[c] 3.3[e] | 0.1[a, c] 0.5[a, e] | Tropical forest, Brazil, background, this work |
| Sep/2012 | 1.4[c] 1.6[e] | 7.0[c] 6.1[e] | 1.4[a, c] 1.2[a, e] | Tropical forest, Brazil, fresh smoke, this work |
| Sep/2012 | 2.4[c] 2.4[e] | 2.3[c] 2.3[e] | 0.1[a, c] 0.3[a, e] | Tropical forest, Brazil, aged smoke, this work |
| Sep/1979 Aug/1980 | 2.4[b] 2.3[c] 0.2[d] | - | - | Grassland/Tropical forest, Brazil, Greenberg and Zimmerman, 1984 |
| Jun/1984 | 2.3[b] | - | - | Tropical forest, Guyana, Gregory et al., 1986 |
| Jul/1985 | 2-4[c] | - | - | Tropical forest, Brazil, Rasmussen & Khalil, 1988 |
| Jul/1985 | ~2[c] | - | - | Tropical forest, Brazil (north of Manaus), Zimmerman et al., 1988 |
| Oct/1995 | - | - | 3-5[d] | Southern Ocean, south of Tasmania, Mauldin et al., 1997 |
| Jul/1996 | 3.1[b] 1.4[c] 0.2[d] | 0.15[b] 0.19[c] 0.48[d] | - | Tropical forest, Peru, Helmig et al., 1998 |
| May/1997 | 1-4[b] | - | 8-13[c, a] | Boreal forest, USA (Sierra Nevada), Dreyfus et al., 2002 |
| Aug/2000 | - | - | ~17[a] | Savanna, South Africa (Timbavati reserve), Hobbs et al., 2003 |
| Jan/2000 | 0.4/0.7/0.5[f] | - | - | Tropical forest, Brazil (Tapajós), Greenberg et al., 2004 |
| Mar/1998 | 1.7/2.9/3.1[f] | - | - | Tropical forest, Brazil (Balbina), Greenberg et al., 2004 |
| Feb/1999 | 6.6/6.9/6.6[f] | - | - | Tropical forest, Brazil (Jaru reserve), Greenberg et al., 2004 |
| Feb/1999 | 2.0/1.3/1.2[f] | - | - | Grassland, Brazil (FNS site), Greenberg et al., 2004 |
| Jul/2001 | 1.1-5.8[c] | < 2[c] / 2-10[d] | 5.5[c] | Tropical forest, Brazil (north of Manaus), Kuhn et al., 2007 |
| Sep/2004 | - | 0.39[b]/0.61[c]/1.2[d] | 0.2[b]-9[c] | Tropical forest, Brazil (north of Manaus), Karl et al., 2007 |
| Oct/2005 | 2.0[c] 0.1[d] | - | 11[c] 5[d] | Tropical forest, Suriname, Lelieveld et al., 2008 |
| Oct/2005 | - | - | 4.4[c] | Pristine forest, Suriname, Kubistin et al., 2010 |
| Mar/2014 | 2[b] | - | 1.0[b] | Tropical Forest (pasture), Brazil (west of Manaus), Liu et al., 2016 |

[a]*Estimated values. Measurements at* [b]*surface,* [c]*boundary layer,* [d]*free troposphere,* [e]*cloud layer.* [f]*Measurements at 9-12h/12-15h/15-18h*



Figure 1. SAMBBA flights tracks according to their original goals as biogenic emissions (green) and biomass burning (red). The black dots indicate the locations of the taking-off and landing airports. The red points depict the fires detected by MODIS onboard AQUA satellite during the SAMBBA campaign from 14 September - 3 October 2012.








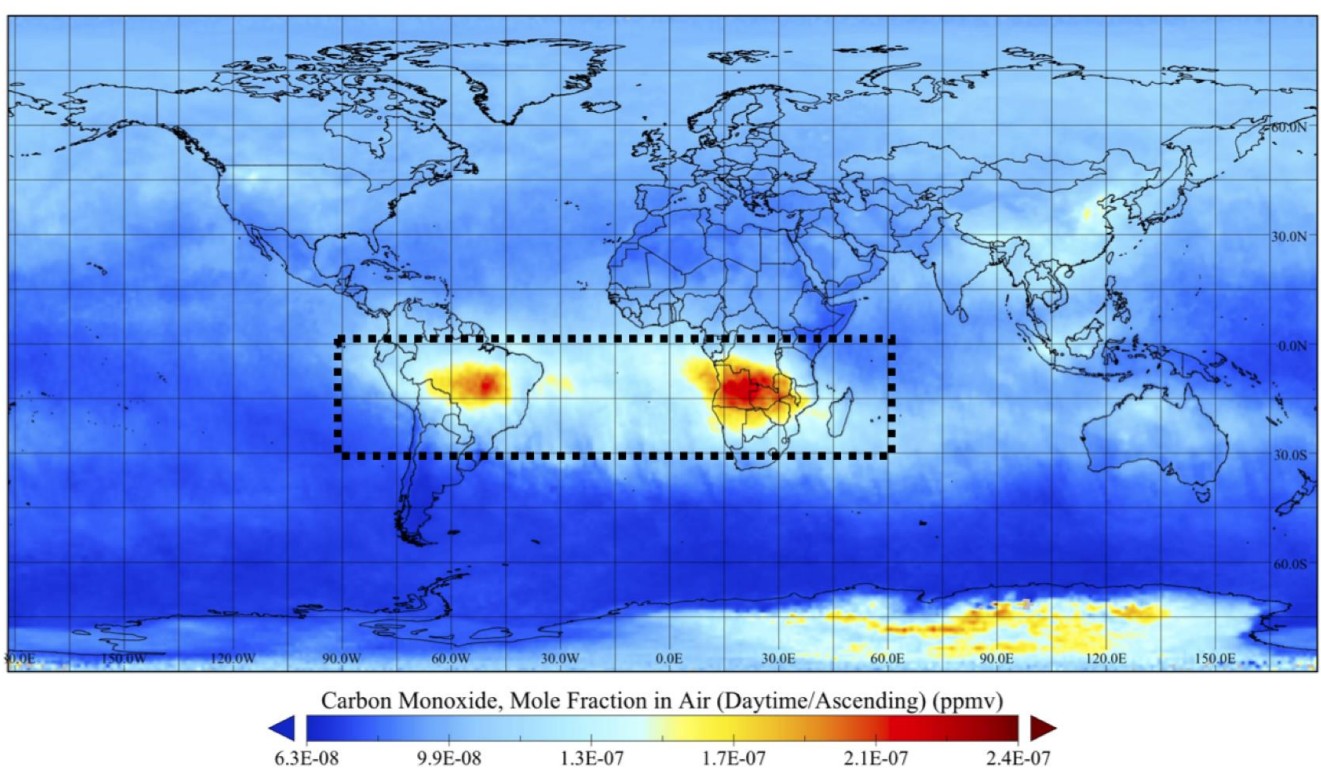

Figure 2. Time-averaged CO (ppmv) over SAMBBA period (14 September - 3 October 2012) from AIRS onboard AQUA satellite during daytime at 500 hPa. The region of interest is indicated on the map.





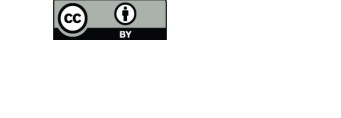


Figure 3. Cross-section of CO (on top), $O_3$ (middle) and $NO_x$ (on bottom) mixing ratios (ppbv) for the three different groups: background environment (on the left), fresh smoke plume (t < 2 hours, on the middle) and aged smoke plume (t > 2 hours, on the right). The aircraft data were interpolated from the various vertical profile measurements using kriging correlation
method. Grey lines show the flight tracks. Hour is presented in local time.



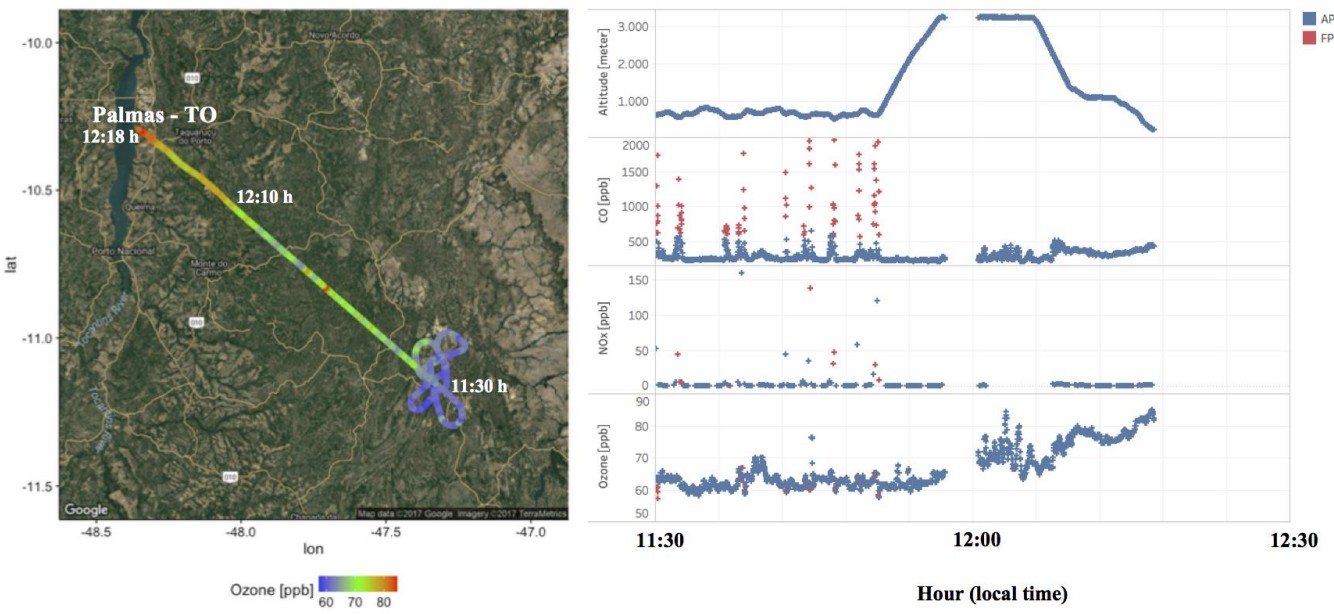

Figure 4. On the left, the track of flight B742 that landed in Palmas – TO. The color bar represents the measured $O_3$ mixing
ratios (ppbv) along the flight track. On the right, from top to bottom, the altitude, and the CO, $NO_x$ and $O_3$ mixing ratios
(ppbv) measured along the B742 flight track. The red and blue dots represent the parts of the flight track classified as fresh
(FP) and aged (AP) smoke plumes, respectively.





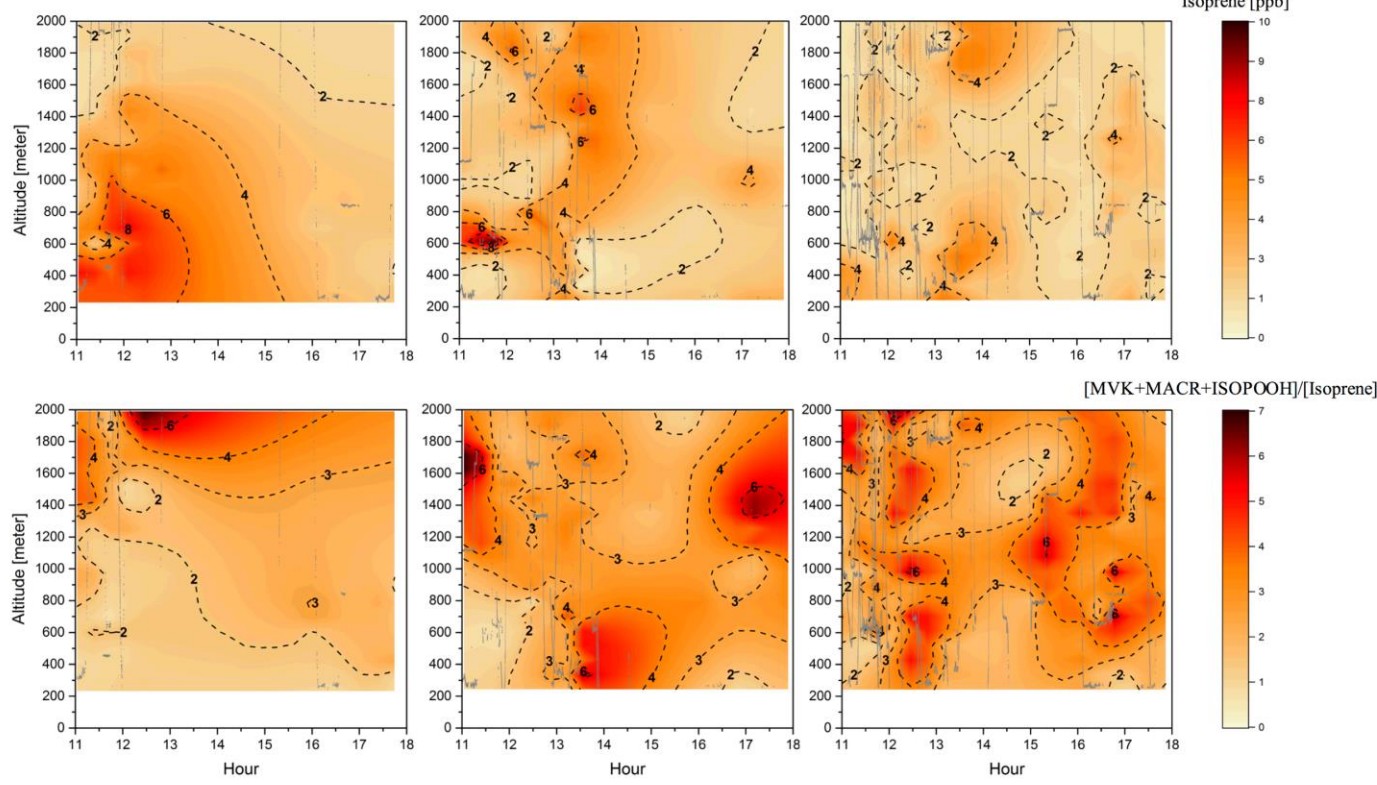

Figure 5. Cross-section of the isoprene mixing ratio (ppbv) (top) and the [MVK+MACR+ISOPOOH]/[Isoprene] ratio (bottom) for the three different groups: background environment (on the left), fresh smoke plume (t < 2 hours, in the middle), and aged smoke plume (t > 2 hours, on the right). The aircraft data were interpolated from the various vertical profile measurements using kriging correlation method. White dashed lines show the flight tracks. Hour is presented in local time.







Figure 6. Isoprene, Acetonitrile and CO mixing ratios (ppbv) as function of daytime (local time) for the different chemical regimes previously classified as background (green dots), fresh smoke plume (red dots), and aged smoke plume (blue dots). Black dashed lines, and the numbers next to them, represent the mean values of the measurements taken below 2,000 m altitude. Hour is presented in local time (11:00 – 18:00 h).





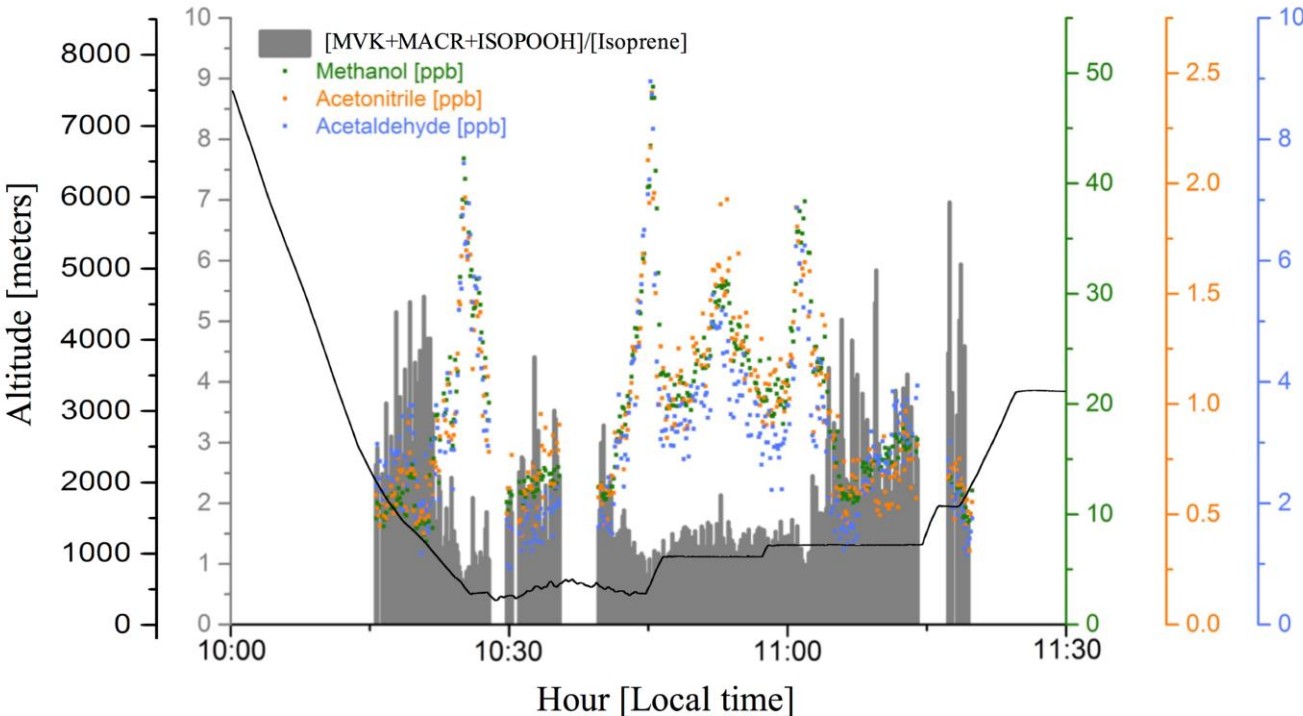

Figure 7. Methanol (green dots), Acetonitrile (orange dots), and Acetaldehyde (blue dots) mixing ratios (ppbv), and the [MVK+MACR+ISOPOOH]/[Isoprene] ratio (gray bars), during a plume interception along the flight track B732 in different altitudes.






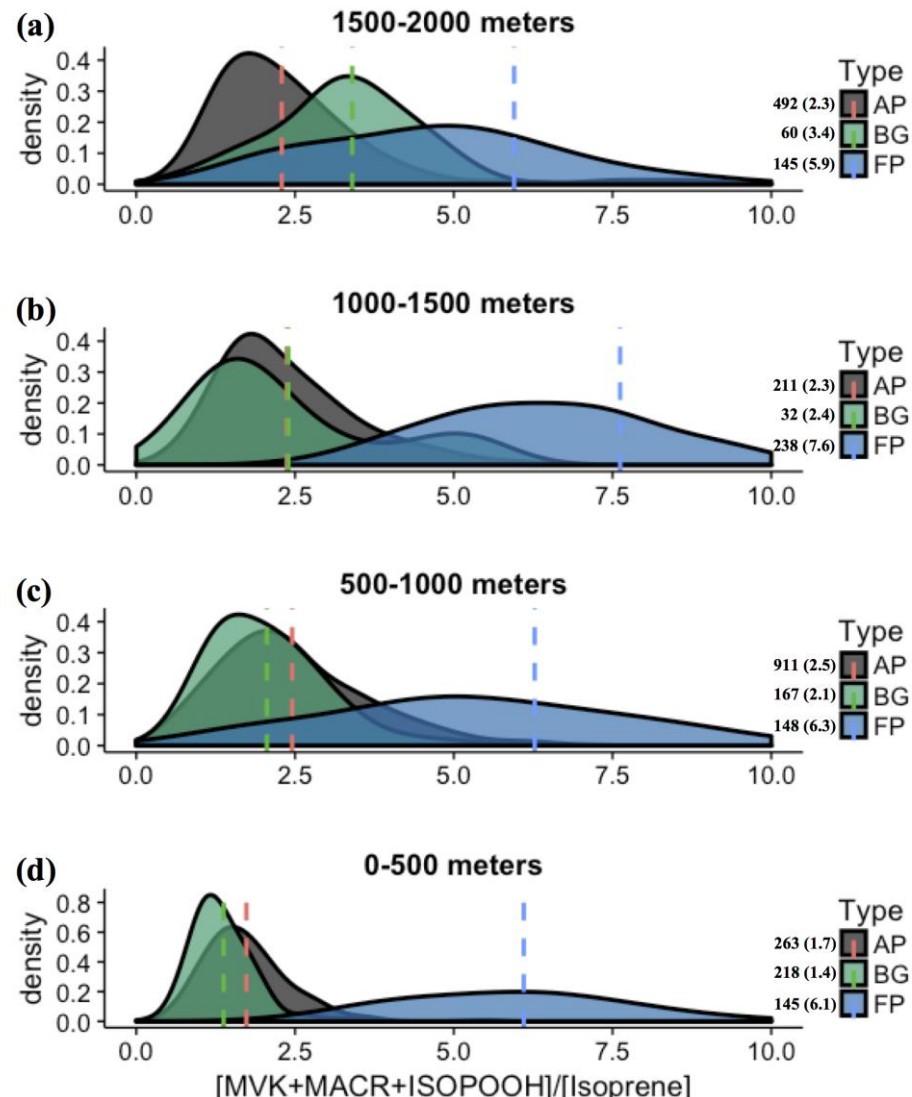

Figure 8. Density distributions of the ratio [MVK+MACR+ISOPOOH]/[isoprene], at the altitude layers (a) 1,500 - 2,000 m, (b) 1,000 - 1,500 m, (c) 500 - 1,000 m and (d) 0 - 500 m. The Kernel analysis was carried out considering the classification for background (BG), aged smoke (AP), and fresh smoke plumes (FP). The number of samples and mean values for each group are depicted near the color bars.




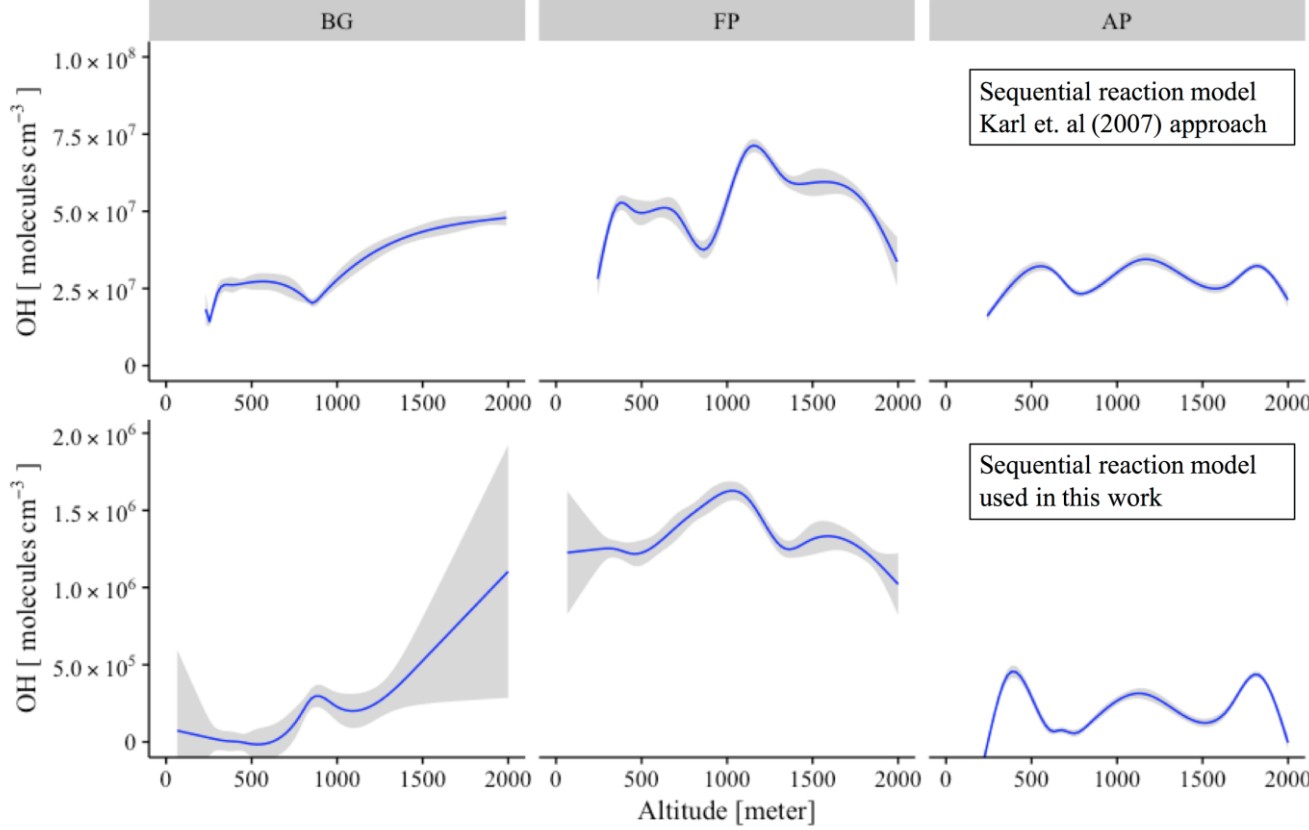

Figure 9. Vertical profile of OH concentration (molecules cm$^{-3}$) for the different chemical regimes: background environment (BG), fresh smoke plume (FP), and aged smoke plume (AP). On top, the sequential reaction model according to the original approach of Karl et al. (2007), and on bottom, the new approach used in this work. Blue lines are the trend lines and grey intervals represents the level of confidence (0.95) used.

