# Peer review of "Biomass burning emissions disturbances on the isoprene oxidation in a tropical forest"

_Atmospheric Chemistry and Physics, 2017_

## Referee Comment (RC1) · Anonymous Referee #1 · 24 Feb 2018

This manuscript examines the oxidative capacity of the atmosphere above the Amazon Rainforest, focusing in particular on variations with altitude within the boundary layer and cloud layer, and on the perturbations to that oxidative capacity wrought by biomass burning plumes. Working with atmospheric measurements conducted from aircraft during the SAMBBA campaign in the Brazilian Amazon in 2012, the authors use CO as a biomass burning tracer to identify plumes, and further use ratios of ozone and CO enhancement to identify aged and fresh plumes. The authors then use the ratio between concentrations of isoprene's reactive products (MVK + MACR + ISOPOOH, measured by PTR-MS) and isoprene itself (measured by PTR-MS and corrected for furan interference by comparison to WAS-GC-FID), along with an estimation of processing time since isoprene emission to the airmass, to calculate concentrations of

[OH] within that airmass. Fresh plumes are found to have substantially higher product-to-isoprene ratios, and higher estimated OH, than aged plumes or background conditions, demonstrating that biomass burning events can intensify oxidation processes. The authors also highlight variations in altitude: in background air masses, calculated [OH] increased markedly from the surface layer to the cloud layer, while the increase was diminished in aged plumes and in fresh plumes it decreased slightly.

Measurements and models of the oxidative capacity of the remote troposphere, and in particular of OH concentrations in forested regions, have been notoriously difficult and error-prone, but their accurate estimation is crucial to understanding the fates of biogenic emissions in those environments and the roles that such biogenic emissions play in OH titration and/or recycling. New or updated methods to make such estimations, such as the one the authors provide here, are therefore an important research goal for atmospheric chemists. However, as explained in the general comments below, more characterization of the discrepancies with alternate methods and of potential biases and errors in this new method are necessary before it can be broadly applied.

General comments:

1) It would greatly enhance the utility and strength of both the methods and main conclusions - the variation in oxidative capacity of the Amazon boundary layer between pristine, fresh, and aged smoke plume conditions - in this manuscript to include more discussion of potential sources of error and uncertainty, and the spread in the data. In particular, the lack of ranges and error bounds on measurements from this study and on the [OH] numbers reported in Table 4 makes them difficult to interpret, because it is not clear (a) whether these results hold broadly for all plumes under all conditions, and for the background under all conditions, and (b) what the uncertainty is in the estimates. There are a lot of assumptions & steps to get from what you measure (which itself has ranges and uncertainties) to what you infer, and I think it's slightly misleading and less useful to report single numbers as averages in the three conditions. Figure 8 does a good job breaking this down to show variation in the product:isoprene ratio within and

across altitudes under the three conditions, but it would be nice to see that translated to uncertainties in OH, and to see an expansion in the discussion of potential sources of error

2) Kind of as a continuation of the previous comment, it would also benefit this manuscript to have more discussion of the departure of your results from those of Karl et al. (2007) and other studies, and whether or not those discrepancies are expected from the differences in methods used between the studies. It is surprising that such similar methods can give such vastly differing results, and merits further description of why the estimates in Karl et al. were two orders of magnitude higher. There is a section listing results of some previous studies (L 388 - 408), but it does not delve far into why different studies found such different values and what reasons exist to believe certain numbers in place of others. In particular, the background values reported in this study seem substantially lower than recent studies that used similar organic ratios (e.g. Liu et al. 2016) - how can these be reconciled?

3) The use of a single value for the total yield of [MVK + MACR + ISOPOOH] from isoprene (0.55) and for the rate of [MVK + MACR + ISOPOOH] + OH (6.1e-11 cm3 molec-1 s-1) seems potentially problematic. Under the range of conditions reported in this study, the pathways of isoprene oxidation can differ drastically, from an overwhelming fraction of the isoprene-derived peroxy radicals reacting with NO (giving combined MVK + MACR yields ∼57%, from the numbers on L 112, and no ISOPOOH) to very low-NO chemistry in which a large fraction isomerizes (forming MVK and MACR in small yields) and the rest reacts with HO2 to form ISOPOOH in very large yields (up to 93%, Liu et al. 2013). Given that the three first-generation products also have very different oxidation rates, as you note in section S2, the rate of product loss can also vary widely depending on the oxidative conditions (particularly the concentrations of NO). For example, the attached figures show the steady-state daytime [ISOPOOH + MVK + MACR] / isoprene ratio, the steady state [ISOPOOH] / [MVK + MACR] ratio, and the steady-state [OH] as a function of [NO] for various isoprene oxidation models

(MCM from Jenkin et al. 2015, Caltech from Wennberg et al. 2018, and GEOS-Chem versions 10 and 11), using the box modeling conditions described in Jenkin et al. 2015. The variations with NO are pronounces.

These variations in yields and rates could be treated in different ways: you could just incorporate them as uncertainties in the model, in which case, they merit much more discussion in this manuscript; or, you could use the measured NO in the air masses sampled to calculate the assumed fractions of MVK, MACR, and ISOPOOH produced from isoprene, and from that to calculate the bulk oxidation rate of these products, using one of the isoprene oxidation models shown in the attached figures. You could also consider bringing temperature into the equation, since all the oxidation rates (and some of the yields) vary with temperature as well.

Content comments:

L 44 - please clarify whether or not this includes methane. L 45 - what is meant by the "balance" of VOC's emitted by anthrop[ogen]ic and biogenic sources? As written, it sounds like the anthropogenic and biogenic sources are in balance with each other, which is presumably not the case; is this meant to imply that sources are balanced with sinks (i.e. oxidation)? L 51 - it is unclear here what is meant by the clause "affect the VOC... process." Does this mean that the Amazon and similar regions have unique oxidative chemistry? The OH radical initiates isoprene oxidation ubiquitously, so I'm not sure how the "OH radical strengthening the process" in the Amazon uniquely affects VOC oxidation. L 89-91 - please include citations for these "significant advances" - as it stands, this sentence is too vague to identify what advances are being referred to. L 91 - what are these two pathways? RO2 and NO? The peroxy radicals produced from isoprene can undergo a number of subsequent reaction pathways, including with NO (e.g. Tuazon et al. 1990), HO2 (e.g. St. Clair et al. 2015), RO2 (e.g. Jenkin et al. 1998), and self-reactions / isomerizations (e.g. Peeters et al. 2009). Therefore it is not just an accurate understanding of two pathways alone that are required! L 111-114 - I see the figure in Jenkin et al. 2015 these are reported from, but it's important

that these are also from a specific environment - it is not just the level of NO that matters for these yields (because other pathways can yield MVK and methacrolein, and because concentrations of other reactants can influence the branching between isoprene peroxy radical isomers that react with NO to yield these products). L 152 - ISOPOOH is not isobaric with MVK and MACR - it's a decomposition interference. If it was only calibrated with MVK and MACR, do you have some estimate of how efficient the decomposition of ISOPOOH –> m/z 71 was? L 217-218 - see general comment (3) above about these assumptions regard the oxidation rates and yields of [MVK + MACR + ISOPOOH] L 303 - what is meant by "producing a value around 50% and 14%, respectively"? Are these the measured reductions in isoprene mixing ratios from the BG to the plumes? L 315 - here you say the [MVK+MACR+ISOPOOH]/isoprene ratio is low in fresh plumes, which appears to be corroborated in Figure 7, but on L 320-321 and later you quote high values of the [MVK+MACR+ISOPOOH]/isoprene ratio in fresh plumes, as shown in Figure 8. How do these coincide? L 336 - what is meant by "as in AP levels"? It appears from Table 4 that AP isoprene oxidation is constant with altitude, unlike BG. L 344-345 - how does an increase in photolysis rates increase the [MVK+MACR+ISOPOOH]/isoprene ratio? If it speeds up oxidation in general, it may remove isoprene faster, but if it also increases the photolysis of MVK, MACR, and ISOPOOH (rather than isoprene) it may cause faster removal of the products instead. L 378 - two orders of magnitude is a lot! The values quoted from here down to L 408 span a wide range, and it is therefore not clear whether saying the "OH concentration values presented in this study agree in order of magnitude with more modeled and observed values previously reported" (L 388-389) is a useful metric. See general comments (1) and (2) above. L 404 - Why is the quoted range from Williams et al. (2001) 0.6-1.1 here but 0.7-101 on L 391 above?

Minor copyediting comments:

L 19 - "trace gases measurements" –> "trace gas measurements" L 45 - "anthropic" –> "anthropogenic" L 46 - "sub-products" - does this just mean later-generation products? Or co-products? Unclear. L 186 - "flights" –> "flight" L 203 - "ISOPOHH" –> "ISOPOOH" L 304 - ">1,2000 m" –> ">1,200 m" L 354 - "showed" –> "shown" L 372 - "value" –> "values" L 379 - "tends" –> "tend" L 396 - "from Sacramento region" –> "from the Sacramento region" L 430 - "regarding the isoprene" –> "regarding isoprene" L 431 - "Near fresh and aged smoke" is not needed L 503 - this citation is missing a journal name L 506 - all of the authors are duplicated in the reference for Feiner, et al. L 634 - the URL isn't needed here L 655 - a number of the authors are duplicated in the reference for Stroud, et al. L 666 - does this reference refer to the chapter in "Atmospheric and Aerosol Chemistry"? If so it should include: "In: McNeill V., Ariya P. (eds) Atmospheric and Aerosol Chemistry. Topics in Current Chemistry, vol 339. DOI https://doi.org/10.1007/128_2012_359" L 671 - Whalley's name is duplicated in the reference for Whalley, et al, 2013. Also, given the title, is this just a duplicate of the Whalley, Stone, and Heard reference above? L 685 - this ACPD reference should probably be for the final ACP paper instead

References:

Jenkin, M. E., Young, J. C. and Rickard, A. R.: The MCM v3.3.1 degradation scheme for isoprene, Atmos. Chem. Phys., 15(20), 11433–11459, doi:10.5194/acp-15-11433-2015, 2015.

Karl, T., Guenther, A., Yokelson, R. J., Greenberg, J., Potosnak, M., Blake, D. R. and Artaxo, P.: The tropical forest and fire emissions experiment: Emission, chemistry, and transport of biogenic volatile organic compounds in the lower atmosphere over Amazonia, J. Geophys. Res., 112(D18), D18302, doi:10.1029/2007JD008539, 2007.

Liu, Y. J.; Herdlinger-Blatt, I.; McKinney, K. A.; Martin, S. T. Production of Methyl Vinyl Ketone and Methacrolein via the Hydroperoxyl Pathway of Isoprene Oxidation. Atmos. Chem. Phys. 2013, 13, 5715−5730.

Liu, Y., Brito, J., Dorris, M., Rivera-Rios, J. C., Seco, R., Bates, K. H., Artaxo, P., Junior, S. D., Keutsch, F., Kim, S., Goldstein, A. H., Guenther, A. B., Manzi, A.,

Souza, R., Springston, S. R., Watson, T. B., McKinney, K. A. and Martin, S. T.: Isoprene Photochemistry over the Amazon Rain Forest, Proc. Natl. Acad. Sci., doi:10.1073/pnas.1524136113, 2016.

Peeters, J.; Nguyen, T. L.; Vereecken, L. HOx Radical Regeneration in the Oxidation of Isoprene. Phys. Chem. Chem. Phys. 2009, 11, 5935−9.

St Clair, J. M.; Rivera-Rios, J. C.; Crounse, J. D.; Knap, H. C.; Bates, K. H.; Teng, A. P.; Jørgensen, S.; Kjaergaard, H. G.; Keutsch, F. N.; Wennberg, P. O. Kinetics and Products of the Reaction of the First-Generation Isoprene Hydroxy Hydroperoxide (ISOPOOH) with OH. J. Phys. Chem. A 2016, 120, 1441−1451.

Tuazon, E. C.; Atkinson, R. A Product Study of the Gas-Phase Reaction of Methacrolein with the OH Radical in the Presence of NOx. Int. J. Chem. Kinet. 1990, 22, 591−602.

Wennberg, P. O., Bates, K. H., Crounse, J. D., Dodson, L. G., McVay, R. C., Mertens, L. A., Nguyen, T. B., Praske, E., Schwantes, R. H., Smarte, M. D., St Clair, J. M., Teng, A. P., Zhang, X., and Seinfeld, J. H. Gas-Phase Reactions of Isoprene and its Major Oxidation Products. Chemical Reviews. 2018, DOI: 10.1021/acs.chemrev.7b00439

[Figure]

[Figure]

ISOPOOH/(MVK+MACR) Ratio vs NO$_x$, 298 K

daytime mean $\chi_{NOx}$ (ppbv)

daytime mean $\chi_{ISOPOOH}/(\chi_{MVK}+\chi_{MACR})$

Reduced
MCM
GC v10
GC v11

---

## Referee Comment (RC2) · Anonymous Referee #2 · 3 Mar 2018

Santos et al. present an informative study of how biomass burning emissions during SAMBBA field campaign changed the oxidative capacity of Amazon rainforest. Fresh plumes especially seem to favor isoprene oxidation compared to aged plumes, which is an important result. But the approach and methods need to be further strengthened. I recommend the following for additional analyses: 1. Section 2.2: Classification method of flight tracks: Since the study focuses on aircraft measurements, it is very important to understand local background is different from regional background. Although CO background for a large region could be around 150 ppb as suggested by the authors, the local background could be changing much more dynamically, since it is influenced by plume. One approach might have been to classify flight tracks over a given region based on O3 or NO thresholds and determine local background CO for these tracks.

Some discussions of local background variations and justification of a constant CO background is needed

2. Table 2: Values of ER (delta ozone/delta CO) vary significantly in different regions. While a threshold of <0.1 as an upper limit may be reasonable for fresh biomass smoke (based on Table 2), using this same threshold as a lower limit for aged smoke is not very convincing. At the minimum, some sensitivity tests are needed where the threshold for aged smoke is increased to say 0.5. How will this change in threshold affect results presented in this study? Also, ozone formation and photochemistry can be slowed down in thick smoke plumes or under cloudy conditions. Is this considered in the ER comparisons?

3. Figure 9: There are large differences between Karl et al. 2007 and the results presented in this study for vertical profiles of OH. Can additional supporting evidence from measurements be provided to show which approach is better? One approach might be to look at trends with altitude.

For example, as altitude increases from 0 to 2000 m, Karl et al. 2007 report OH increases by a factor of 2 (2.5 e7 to 5 e7). But the authors report a much larger increase (close to zero at 500 m to 1e6 at 2000 m). Did the authors observe large increases in ozone and NO with altitude similar to OH increase? In other words, even if quantitative OH measurements may not be available, some analysis about predicted OH trends with altitude and whether these can be justified based on other measurements like ozone and NO could be provided.

Line 355: 360: Fresh biomass plumes could be expected to be low NOx since VOC/NOx ratio may be very high in these plumes, even though fires also emit NOx. Could the authors comment on how the ratio of isoprene oxidation products to isoprene would vary depending on whether NOx regimes are low or high in plumes, and also how these NOx regimes would differ in the local background?
* * *
2017.

---

## Referee Comment (RC3) · Anonymous Referee #3 · 14 Mar 2018

General This is an interesting study on how biomass burning emissions interact with isoprene (ISO) oxidation under the conditions of a tropical forest studied in the SAMBBA campaign. The paper applies an interesting air mass classification. The OH estimation by an analytic experssion shoul dbe handled with care. Referencing should be updated through the paper. The real structures / names of 'ISOPOOH' should at least be given once in the manuscript. Maybe the use of abbreviations can be reduced as some sentences become difficult to understand.

How big is the identied increase in oxidation capacity really ? Please consider possible uncertainties regarding the analytical OH calculation.

Overall, this paper should be acceptable for ACP subject to revisions somewhere between minor and major.

[Figure]

Details

Abstract: Maybe the increase of oxidation capacity dowwind a BB event can be more clearly put into the abstract ?

Line 29ff : Edit the sentence 'The oxidation of isoprene is higher....'

Line 70ff: It strikes me that this part does not make any reference to the important Caltech studies starting with Paulot et al.

Line 146 ff: How do the authors assure that ISOPOOH does contribute to the measured m/z = 73 at all ? Has there been any test for this ? How is the inlet system designed to allow measurement of this species ?

Line 200 ff: Isn't such quasi-analytical approach prone to errors ? How do OH levels calculated by eqn (2) compare to detailed model results ? Can this be compared to establish the validity of eqn (2) ? How uncertain are the OH concentrations calculated ? Is there any chance to compare the analytical results with, say, a 1-D or a box-trajectory model result ?

Line 203: ISOPOOH , not ISOPOHH

Line 363: Maybe I oversee something but what exactly is that 'Sequential reaction approach' ?

---

## Author Comment (AC2) · 14 May 2018

Dear Referee#1,

Thank you for your criticisms and suggestions to the manuscript. Most of the modifications were made in the manuscript (attached) and below are the comments to the questions made previously.
* * *
Referee comment: 1. Section 2.2: Classification method of flight tracks: Since the study focuses on aircraft measurements, it is very important to understand local background is different from regional background. Although CO background for a large region could be around 150 ppb as suggested by the authors, the local background

could be changing much more dynamically, since it is influenced by plume. One approach might have been to classify flight tracks over a given region based on O3 or NO thresholds and determine local background CO for these tracks. Some discussions of local background variations and justification of a constant CO background is needed

Author comment: The flights were classified according to their scientific objectives as either biogenic or biomass burning flights. Additionally, we only considered the data collected below 2,000 m and between 11:00 am and 6:00 pm to capture the difference in the oxidative capacity activity along the planetary boundary layer (PBL) during daytime, since the OH concentration is regulated by photochemistry. Some flights, despite the classification in the planning phase, had parts of their tracks passing through unpolluted regions, smoke haze, or even interception of fresh smoke plumes. An alternative would be the geographic classification of flights in different groups and for each group having a background value, but the insufficient number of flights (n) for some regions, especially the cleaner regions, restricted the use of such method. According to the literature, the Amazon rainforest atmosphere has a background level of CO mixing ratio typically around 100 ppbv. However, the mean CO inflow into the Amazon basin during the SAMBBA period at 500 hPa, retrieved from Atmospheric Infrared Sounder (AIRS) measurements onboard the AQUA satellite, ranged between 140 and 160 ppbv (Figure 1.a). Figure 1.b shows that this hemispheric inflow is quite homogeneous along the vertical column up to around 400 hPa and in fact, there were few SAMBBA samples with CO mixing ratio values ~100 ppbv. The histogram (Figure 2) also present the frequency distribution for all SAMBBA flights, with the value of 150 ppbv as an upper limit. Therefore, in this case, we adopted a threshold of 150 ppbv to represent the background of CO in the Amazon atmosphere during SAMBBA campaign.

Referee comment: 2. Table 2: Values of ER (delta ozone/delta CO) vary significantly in different regions. While a threshold of <0.1 as an upper limit may be reasonable for fresh biomass smoke (based on Table 2), using this same threshold as a lower limit for aged smoke is not very convincing. At the minimum, some sensitivity tests

are needed where the threshold for aged smoke is increased to say 0.5. How will this change in threshold affect results presented in this study? Also, ozone formation and photochemistry can be slowed down in thick smoke plumes or under cloudy conditions. Is this considered in the ER comparisons?

Author comment: Based in the study of Mauzerall et al., 1998, we choose the value ER = 0.1 as a representative value of ∼2 hours plume age. In fact, the ER data (delta ozone/delta CO) are scarce in the literature and we had difficulties in getting them together as an organized table (Table 2). All the factors mentioned by the referre #2 can impact the ER values, although we had worked with the definition of fresh/recent plume from Mauzerall et al., 1998 and the available observations of plume age in tropical and subtropical sites.
* * *
Referee comment: 3. Figure 9: There are large differences between Karl et al. 2007 and the results presented in this study for vertical profiles of OH. Can additional supporting evidence from measurements be provided to show which approach is better? One approach might be to look at trends with altitude. For example, as altitude increases from 0 to 2000 m, Karl et al. 2007 report OH increases by a factor of 2 (2.5 e7 to 5 e7). But the authors report a much larger increase (close to zero at 500 m to 1e6 at 2000 m). Did the authors observe large increases in ozone and NO with altitude similar to OH increase? In other words, even if quantitative OH measurements may not be available, some analysis about predicted OH trends with altitude and whether these can be justified based on other measurements like ozone and NO could be provided.

Author comment: Unfortunately, we did not have a direct measurement of [OH] during SAMBBA campaign and our comparison is based in absolute [OH] values, as you can see in the literature presented in the manuscript. Our conclusions are based on the absolute value of [OH], calculated using the new plume age methodology and comparing with recent GoAmazon campaign. We also added

some comments in the manuscript and in the section S5 (attached) we added the vertical profile for [NOx] and [O3] mixing ratios during SAMBBA campaign.

———————————————————————————————————————————

Referee comment: Line 355: 360: Fresh biomass plumes could be expected to be low NOx since VOC/NOx ratio may be very high in these plumes, even though fires also emit NOx. Could the authors comment on how the ratio of isoprene oxidation products to isoprene would vary depending on whether NOx regimes are low or high in plumes, and also how these NOx regimes would differ in the local background?

Author comment: We detected the highest values for NOx in the fresh and aged smoke plumes ($\sim$850 m) and the ratio [MVK+MACR+ISOPOOH]/[Isoprene] also presented a strong enhancement from 250 to 1250 m. These results go towards to the reaction of isoprene with peroxy radicals (HO2) as an alternative pathway to produce OH in an unpolluted environment.

Please also note the supplement to this comment: https://www.atmos-chem-phys-discuss.net/acp-2017-1083/acp-2017-1083-AC2-supplement.zip

————————————————————————

[Figure]

**Fig. 1.** Time averaged CO (ppmv) over SAMBBA period from AIRS onboard AQUA satellite during daytime: (a) global map at 500 hPa, and (b) cross section of longitude-pressure within the region on the map.

**Histogram - SAMBBA Flights**

**Fig. 2.** Histogram that present the frequency distribution of CO [ppbv] for all SAMBBA flights in Amazon rainforest.

---

## Author Response (AR1)

This manuscript examines the oxidative capacity of the atmosphere above the Amazon Rainforest, focusing in particular on variations with altitude within the boundary layer and cloud layer, and on the perturbations to that oxidative capacity wrought by biomass burning plumes. Working with atmospheric measurements conducted from aircraft during the SAMBBA campaign in the Brazilian Amazon in 2012, the authors use CO as a biomass burning tracer to identify plumes, and further use ratios of ozone and CO enhancement to identify aged and fresh plumes. The authors then use the ratio between concentrations of isoprene's reactive products (MVK + MACR + ISOPOOH, measured by PTR-MS) and isoprene itself (measured by PTR-MS and corrected for furan interference by comparison to WAS-GC-FID), along with an estimation of processing time since isoprene emission to the airmass, to calculate concentrations of [OH] within that airmass. Fresh plumes are found to have substantially higher product-to-isoprene ratios, and higher estimated OH, than aged plumes or background conditions, demonstrating that biomass burning events can intensify oxidation processes. The authors also highlight variations in altitude: in background air masses, calculated [OH] increased markedly from the surface layer to the cloud layer, while the increase was diminished in aged plumes and in fresh plumes it decreased slightly.

Measurements and models of the oxidative capacity of the remote troposphere, and in particular of OH concentrations in forested regions, have been notoriously difficult and error-prone, but their accurate estimation is crucial to understanding the fates of biogenic emissions in those environments and the roles that such biogenic emissions play in OH titration and/or recycling. New or updated methods to make such estimations, such as the one the authors provide here, are therefore an important research goal for atmospheric chemists. However, as explained in the general comments below, more characterization of the discrepancies with alternate methods and of **potential biases and errors in this new method are necessary** before it can be broadly applied.

General comments:

1) It would greatly enhance the utility and strength of both the methods and main conclusions - the variation in oxidative capacity of the Amazon boundary layer between pristine, fresh, and aged smoke plume conditions - **in this manuscript to include more discussion of potential sources of error and uncertainty, and the spread in the data**. In particular, the lack of ranges and error bounds on measurements from this study and on the [OH] numbers reported in Table 4 makes them difficult to interpret, because it is not clear (a) whether these results hold broadly for all plumes under all conditions, and for the background under all conditions, and (b) what the uncertainty is in the estimates. There are a lot of assumptions & steps to get from what you measure (which itself has ranges and uncertainties) to what you infer, and I think it's slightly misleading and less useful to report single numbers as averages in the three conditions.

(1) Following the reviewer's suggestion standard errors were included in the values reported here (including Table 4). Although the Table 4 has been created to compare the measurements and estimated values from this study with the literature, we also recognize that there is a lack of information level between our study and other measurements in Amazon rainforest. We push forward the details about Isoprene, its oxidation products and OH, using the information about altitude level (surface, boundary layer, free troposphere, cloud layer) and the atmospheric condition (background, fresh and aged smoke). Unfortunately, we do not have the same details in the literature as we have in our study.

Figure 8 does a good job breaking this down to show variation in the product:isoprene ratio within and across altitudes under the three conditions, but it would be nice to see that translated to uncertainties in OH, and to see an expansion in the discussion of potential sources of error

> Yes, we agree with the reviewer. The OH uncertainty and potential sources of error has been included in the last paragraph in the section 3.3. In Figure 8, the distribution density and its average value, together with number of samples considered, give us a good overview of the data. The average values can simplify the comparison process, but the kernel density distribution show how dispersed can be the measurements.

2) Kind of as a continuation of the previous comment, it would also benefit this manuscript to have **more discussion of the departure of your results from those of Karl et al. (2007) and other studies, and whether or not those discrepancies are expected from the differences in methods used between the studies**. It is surprising that such **similar methods can give such vastly differing results, and merits further description of why the estimates in Karl et al. were two orders of magnitude higher.** There is a section listing results of some previous studies (L 388 - 408), but it does not delve far into why different studies found such different values and what reasons exist to believe certain numbers in place of others. In particular, the background values reported in this study seem substantially lower than recent studies that used similar organic ratios (e.g. Liu et al. 2016) - how can these be reconciled?

> (2) As mentioned in the manuscript (section 2.3), we modified the processing time *t* in the sequential reaction model to represent not only the vertical transport but also the horizontal atmospheric circulation. The differences in the OH calculated through the two versions of sequential reaction model is due to the attempt to improve the transport time along the atmosphere.
>
> The background measurements in this study comes from the classification adopted: specific altitude range (below 2,000 m), time of the day (between 11:00 am and 6:00 pm) and CO levels (threshold of 150 ppbv). These criteria can bring to the result different values. Also, the background value should be understood as a representative value. The Amazon rainforest atmosphere has a background CO mixing ratio typically around 100 ppbv. However, the mean CO inflow into the Amazon Basin throughout the SAMBBA period was approximately 140-160 ppbv. An alternative would be the geographic classification of flights in different groups and for each group having a background value, but the insufficient number of flights *(n)* for some regions, especially the cleaner regions, restricted the use of such method.

3) The use of a single value for the total yield of [MVK + MACR + ISOPOOH] from isoprene (0.55) and for the rate of [MVK + MACR + ISOPOOH] + OH (6.1e-11 cm3 molec-1 s-1) seems potentially problematic. Under the range of conditions reported in this study, the pathways of isoprene oxidation can differ drastically, from an overwhelming fraction of the isoprene-derived peroxy radicals reacting with NO (giving combined MVK + MACR yields ∼57%, from the numbers on L 112, and no ISOPOOH) to very low-NO chemistry in which a large fraction isomerizes (forming MVK and MACR in small yields) and the rest reacts with HO2 to form ISOPOOH in very large yields (up to 93%, Liu et al. 2013). Given that the three first-generation products also have very different oxidation rates, as you note in section S2, the rate of product loss can also vary widely depending on the oxidative conditions (particularly the concentrations of NO). For example, the attached figures show the steady-state daytime [ISOPOOH + MVK + MACR] / isoprene ratio, the steady state [ISOPOOH] / [MVK + MACR] ratio, and the steady-state [OH] as a function of [NO] for various isoprene oxidation models (MCM from Jenkin et al. 2015, Caltech from Wennberg et al. 2018, and GEOS-Chem versions 10 and 11), using the box modeling conditions described in Jenkin et al. 2015. The variations with NO are pronounces.

**These variations in yields and rates could be treated in different ways: you could just incorporate them as uncertainties in the model, in which case, they merit much more discussion in this manuscript**; or, you could use the measured NO in the air masses sampled to calculate the assumed fractions of MVK, MACR, and ISOPOOH produced from isoprene, and from that to calculate the bulk oxidation rate of these products, using one of the isoprene oxidation models shown in the attached figures. **You could also consider bringing temperature into the equation, since all the oxidation rates (and some of the yields) vary with temperature as well.**

> (3) We agree with the reviewer on the impact of different reaction pathways may have

on the OH estimate provide here and have added a discussion accordingly. We find extremely interesting the suggestion of the reviewer; however, we feel that such modification fall outside the scope of the current manuscript, which aimed at studying the impact on OH estimates by Karl et al., (2007) equations through a much improved estimate of transport time. We hope in the near future though to exploit the different reactions pathway might have on the OH estimates accordingly.

Content comments:

L 44 - please clarify whether or not this includes methane. L 45 - what is meant by the "balance" of VOC's emitted by anthrop[ogen]ic and biogenic sources? As written, it sounds like the anthropogenic and biogenic sources are in balance with each other, which is presumably not the case; is this meant to imply that sources are balanced with sinks (i.e. oxidation)? L 51 - it is unclear here what is meant by the clause "affect the VOC... process." Does this mean that the Amazon and similar regions have unique oxidative chemistry? The OH radical initiates isoprene oxidation ubiquitously, so I'm not sure how the "OH radical strengthening the process" in the Amazon uniquely affects VOC oxidation. L 89-91 - please include citations for these "significant advances" - as it stands, this sentence is too vague to identify what advances are being referred to. L 91 - what are these two pathways? RO2 and NO? The peroxy radicals produced from isoprene can undergo a number of subsequent reaction pathways, including with NO (e.g. Tuazon et al. 1990), HO2 (e.g. St. Clair et al. 2015), RO2 (e.g. Jenkin et     al. 1998), and self-reactions / isomerizations (e.g. Peeters et al. 2009). Therefore it   is not just an accurate understanding of two pathways alone that are required! L 111-114 - I see the figure in Jenkin et al.  2015 these are reported from, but it's important that these are also from a specific environment - it is not just the level of NO that matters for these yields (because other pathways can yield MVK and methacrolein, and because concentrations of other reactants can influence the branching between isoprene peroxy radical isomers that react with NO to yield these products). L 152 - ISOPOOH is not isobaric with MVK and MACR - it's a decomposition interference. If it was only calibrated with MVK and MACR, do you have some estimate of how efficient the decomposition of ISOPOOH –> m/z 71 was?

Content comments:

According to Rivera-Rios et al., 2014, the conversion yields of ISOPOOH into MVK and MACR was observed to be greater than 70%, but the decomposition is known to be highly sensitive to instrumental settings such as temperature, contact time and type of surface materials, especially transition metal surfaces (Liu et al., 2013; Nguyen et al., 2014; Rivera-Rios et al., 2014, Liu et al., 2016, Bernhammer et al., 2017).

Liu, Y. J., Herdlinger-Blatt, I., McKinney, K. A. and Martin, S. T.: Production of methyl vinyl ketone and methacrolein via the hydroperoxyl pathway of isoprene oxidation, Atmos. Chem. Phys., 13(11), 5715–5730, doi:10.5194/acp-13-5715-2013, 2013.

Nguyen, T. B., Crounse, J. D., Schwantes, R. H., Teng, A. P., Bates, K. H., Zhang, X., St Clair, J. M., Brune, W. H., Tyndall, G. S., Keutsch, F. N., Seinfeld, J. H. and Wennberg, P. O.: Overview of the Focused Isoprene eXperiment at the California Institute of Technology (FIXCIT): mechanistic chamber studies on the oxidation of biogenic compounds, Atmos. Chem. Phys., 14(24), 13531–13549, doi:10.5194/acp-14-13531-2014, 2014.

Rivera-Rios, J. C., Nguyen, T. B., Crounse, J. D., Jud, W., Clair, J. M. S., Mikoviny, T., Gilman, J. B., Lerner, B. M., Kaiser, J. B., Gouw, J., Wisthaler, A., Hansel, A., Wennberg, P. O., Seinfeld, J. H. and Keutsch, F. N.: Conversion of hydroperoxides to carbonyls in field and laboratory instrumentation: Observational bias in diagnosing pristine versus anthropogenically controlled atmospheric chemistry,, 1–7, doi:10.1002/(ISSN)1944-8007, 2014.

Liu, Y., Brito, J., Dorris, M. R., Rivera-Rios, J. C., Seco, R., Bates, K. H., Artaxo, P., Duvoisin, S., Keutsch, F. N., Kim, S., Goldstein, A. H., Guenther, A. B., Manzi, A. O., Souza, R. A. F., Springston, S. R., Watson, T. B., McKinney, K. A. and Martin, S. T.: Isoprene photochemistry over the Amazon rainforest, Proc. Natl. Acad. Sci. U.S.A., 113(22), 6125–6130, doi:10.1073/pnas.1524136113, 2016.

Bernhammer, A.-K., Breitenlechner, M., Keutsch, F. N. and Hansel, A.: Technical note: Conversion of isoprene hydroxy hydroperoxides (ISOPOOHs) on metal environmental simulation chamber walls, Atmos. Chem. Phys., 17(6), 4053–4062, doi:10.5194/acp-17-4053-2017, 2017.

L 217-218 - see general comment (3) above about these assumptions regard the oxidation rates and yields of [MVK + MACR + ISOPOOH] L 303 - what is meant by "producing a value around 50% and 14%, respectively"? Are these the measured reductions in isoprene mixing ratios from the BG to the plumes? L 315 - here you say the [MVK+MACR+ISOPOOH]/isoprene ratio is low in fresh plumes, which appears to be corroborated in Figure 7, but on L 320- 321 and later you quote high values of the [MVK+MACR+ISOPOOH]/isoprene ratio in fresh plumes, as shown in Figure 8. How do these coincide?

Content comments:
We appreciate this question that help us to avoid a mistake in the manuscript. There is not a classification in Figure 7 about fresh or aged plumes. The Figure 7 only presents the plume interception during the flight B732 (10:00 - 11:30 am) and is possible to observe the different altitude interceptions through the biomass burning tracers. For ~2 hours as a threshold to differentiate fresh to aged plume, the Figure 8 represent well our results.

L 336 - what is meant by "as in AP levels"? It appears from Table 4 that AP isoprene oxidation is constant with altitude, unlike BG. L 344-345 - how does an increase in photolysis rates increase the [MVK+MACR+ISOPOOH]/isoprene ratio? If it speeds up oxidation in general, it may remove isoprene faster, but if it also increases the photolysis of MVK, MACR, and ISOPOOH (rather than isoprene) it may cause faster removal of the products instead.

Content comments:
According to Apel et al., 2002, the high value for $k_{OH}$ is responsible for the majority of the chemical processing of isoprene by OH. As the rate constant of OH with MVK and MACR are lower than isoprene-OH, we expect an increase in the ratio [MVK+MACR+ISOPOOH]/isoprene, especially in polluted environment. The point has been clarified in L.340

Apel, E. C., Riemer, D. D., Hills, A., Baugh, W., Orlando, J., Faloona, I., Tan, D., Brune, W., Lamb, B., Westberg, H., Carroll, M. A., Thornberry, T. and Geron, C. D.: Measurement and interpretation of isoprene fluxes and isoprene, methacrolein, and methyl vinyl ketone mixing ratios at the PROPHET site during the 1998 Intensive, Journal of Geophysical Research: Atmospheres, 107(D3), 7–15, doi:10.1029/2000JD000225, 2002.

L 378 - two orders of magnitude is a lot! The values quoted from here down to L 408 span a wide range, and it is therefore not clear whether saying the "OH concentration values presented in this study agree in order of magnitude with more modeled and observed values previously reported" (L 388-389) is a useful metric. See general comments (1) and (2) above.

Content comments:
The average OH mixing ratio in boundary layer and cloud layer were reported in this study (table 4), with the values varying from 0.1 to 7 x 10$^6$ $molec.\ cm^{-3}$. Most of the studies mentioned don't have a clear distinction in which altitude the values were obtained (boundary layer and cloud layer) and also very important, no distinction in which ambient condition was reported (biogenic environment, fresh or aged smoke plume). The measurement of the OH in the atmosphere still remain controversial as described in the manuscript, even so, the average values reported in table 4 are in the interval of the majority of the references in the manuscript.

L 404 - Why is the quoted range from Williams et al. (2001) 0.6-1.1 here but 0.7-101 on L 391 above?

Minor copyediting comments:

L 19 - "trace gases measurements" –> "trace gas measurements" L 45 - "anthropic" –> "anthropogenic" L 46 - "sub-products" - does this just mean later-generation products? Or co-products? Unclear. L 186 - "flights" –> "flight" L 203 - "ISOPOHH" –> "ISOPOOH" L 304 - ">1,2000 m" –> ">1,200 m" L 354 - "showed" –> "shown" L 372 - "value" –> "values" L 379 - "tends" –> "tend" L 396 - "from Sacramento region" –> "from the Sacramento region" L 430 - "regarding the isoprene" –> "regarding isoprene" L 431 - "Near fresh and aged smoke" is not needed L 503 - this citation is missing a journal name L 506 - all of the authors are duplicated in the reference for Feiner, et al. L 634 - the URL isn't needed here L 655 - a number of the authors are duplicated

in the reference for Stroud, et al. L 666 - does this reference refer to the chapter in "Atmospheric and Aerosol Chemistry"? If so it should include: "In: McNeill V., Ariya P. (eds) Atmospheric and Aerosol Chemistry. Topics in Current Chemistry, vol 339. DOI https://doi.org/10.1007/128_2012_359" L 671 - Whalley's name is duplicated in the reference for Whalley, et al, 2013. Also, given the title, is this just a duplicate of the Whalley, Stone, and Heard reference above? L 685 - this ACPD reference should probably be for the final ACP paper instead

The flights were classified according to their scientific objectives as either biogenic or biomass burning flights. Additionally, we only considered the data collected below 2,000 m and between 11:00 am and 6:00 pm to capture the difference in the oxidative capacity activity along the planetary boundary layer (PBL) during daytime, since the OH concentration is regulated by photochemistry. Some flights, despite the classification in the planning phase, had parts of their tracks passing through unpolluted regions, smoke haze, or even interception of fresh smoke plumes. An alternative would be the geographic classification of flights in different groups and for each group having a background value, but the insufficient number of flights *(n)* for some regions, especially the cleaner regions, restricted the use of such method.

According to the literature, the Amazon rainforest atmosphere has a background level of CO mixing ratio typically around 100 ppbv. However, the mean CO inflow into the Amazon basin during the SAMBBA period at 500 hPa, retrieved from Atmospheric Infrared Sounder (AIRS) measurements onboard the AQUA satellite, ranged between 140 and 160 ppbv (Figure 3.21.a). Figure 3.21.b shows that this hemispheric inflow is quite homogeneous along the vertical column up to around 400 hPa and in fact, there were few SAMBBA samples with CO mixing ratio values ~100 ppbv. The histogram also present the frequency distribution for all SAMBBA flights, with the value of 150 ppbv as an upper limit. Therefore, in this case, we adopted a **threshold** of 150 ppbv to represent the background of CO in the Amazon atmosphere during SAMBBA campaign.

Figure 3.21 Time averaged CO (ppmv) over SAMBBA period (14th of September - 3rd of October 2012) from AIRS onboard AQUA satellite during daytime: (a) global map at 500 hPa, and (b) cross section of longitude-pressure within the region indicated on the map on top.

[Figure]

[Figure]

2.          Table 2: Values of ER (delta ozone/delta CO) vary significantly in different regions. While a threshold of <0.1 as an upper limit may be reasonable for fresh biomass smoke (based on Table 2), using this same threshold as a lower limit for aged smoke is not very convincing. At the minimum, some sensitivity tests are needed where the threshold for aged smoke is increased to say 0.5. How will this change in threshold affect results presented in this study? Also, ozone formation and photochemistry can be slowed down in thick smoke plumes or under cloudy conditions. Is this considered in the ER comparisons?

Based in the study of Mauzerall et al., 1998, we choose the value ER = 0.1 as a representative value of ~2 hours plume age. In fact, the ER data (delta ozone/delta CO) are scarce in the literature and we had difficulties in getting them together as an organized table (Table 2). All the factors mentioned by the referre #2 can impact the ER values, although we had worked with the definition of fresh/recent plume from Mauzerall et al., 1998 and the available observations of plume age in tropical and subtropical sites.

3.        Figure 9: There are large differences between Karl et al. 2007 and the results presented in this study for vertical profiles of OH. Can additional supporting evidence from measurements be provided to show which approach is better? One approach might be to look at trends with altitude.

For example, as altitude increases from 0 to 2000 m, Karl et al. 2007 report OH increases by a factor of 2 (2.5 e7 to 5 e7). But the authors report a much larger increase (close to zero at 500 m to 1e6 at 2000 m). Did the authors observe large increases in ozone and NO with altitude similar to OH increase? In other words, even if quantitative OH measurements may not be available, some analysis about predicted OH trends with altitude and whether these can be justified based on other measurements like ozone and NO could be provided.

Unfortunately, we didn't have a direct measurement of [OH] during SAMBBA campaign and our comparison is based in absolute [OH] values, as you can see in the literature presented in the manuscript. Our conclusions are based on the absolute value of [OH], calculated using the new plume age methodology and comparing with recent GoAmazon campaign. We also added some comments in the manuscript and in the section *S5* we added the vertical profile for $[NO_x]$ and $[O_3]$ mixing ratios during SAMBBA campaign.

Line 355: 360: Fresh biomass plumes could be expected to be low NOx since VOC/NOx ratio may be very high in these plumes, even though fires also emit NOx. Could the authors comment on how the ratio of isoprene oxidation products to isoprene would vary depending on whether NOx regimes are low or high in plumes, and also how these NOx regimes would differ in the local background?

We detected the highest values for NOx in the fresh and aged smoke plumes (~850 m) and the ratio [MVK+MACR+ISOPOOH]/[Isoprene] also presented a strong enhancement from 250 to 1250 m. These results go towards to the reaction of isoprene with peroxy radicals $(HO_2)$ as an alternative pathway to produce OH in an unpolluted environment.

Atmos. Chem. Phys. Discuss.,
https://doi.org/10.5194/acp-2017-1083-RC3, 2018

[Figure]

General This is an interesting study on how biomass burning emissions interact with isoprene (ISO) oxidation under the conditions of a tropical forest studied in the SAMBBA campaign. The paper applies an interesting air mass classification. The OH estimation by an analytic experssion shoul dbe handled with care. Referencing should be updated through the paper. The real structures / names of 'ISOPOOH' should at least be given once in the manuscript. Maybe the use of abbreviations can be reduced as some sentences become difficult to understand.

How big is the identied increase in oxidation capacity really? Please consider possible uncertainties regarding the analytical OH calculation.

As we try to express in the manuscript, there is a controversial discussion about the impact on the oxidative capacity in forest sites. Observational studies conducted in pristine rainforests showing low-NO and high isoprene have consistently reported unaccountably high OH levels, e.g. (Whalley et al., 2011). Rohrer et al. (2014) compiled several previous OH observations in environments characterized by large VOC concentrations, such as forested areas, and concluded that it requires a substantial OH recycling mechanism to reconcile the discrepancy between observations and model outcomes based on the conventional understanding of isoprene photo-oxidation (Logan et al., 1981). However, a different school of thought considers these discrepancies between model and observation of OH production due to instrument artifacts. Mao et al. (2012) directly demonstrated the magnitude of potential instrument artifacts by adapting a novel background characterization method called a chemical removal technique, a method to measure OH in parallel with the traditional Fluorescence Assay with Gas Expansion (FAGE). The study also illustrated that the application of the chemical removal technique results in agreement between observed and model-calculated diurnal OH variations based on the conventional isoprene photo-oxidation.

Our study is based on the premise that different environments, clean and from biomass burning, have an influence on the oxidative capacity. Due this, we used the [MVK+MACR+ ISOPOOH]/[Isoprene] ratio and the hydroxyl radical (OH) indirect calculation to assess the oxidative capacity of the Amazon forest atmosphere in the background, fresh and aged smoke plumes.

Whalley, L. K., Edwards, P. M., Furneaux, K. L., Goddard, A., Ingham, T., Evans, M. J., Stone, D., Hopkins, J. R., Jones, C. E., Karunaharan, A., Lee, J. D., Lewis, A. C., Monks, P. S., Moller, S. J. and Heard, D. E.: Quantifying the magnitude of a missing hydroxyl radical source in a tropical rainforest, Atmos. Chem. Phys., 11(14), 7223–7233, doi:10.5194/acp-11-7223-2011, 2011.

Rohrer, F., Lu, K., Hofzumahaus, A., Bohn, B., Brauers, T., Chang, C.-C., Fuchs, H., Haeseler, R., Holland, F., Hu, M., Kita, K., Kondo, Y., Li, X., Lou, S., Oebel, A., Shao, M., Zeng, L., Zhu, T., Zhang, Y. and Wahner, A.: Maximum efficiency in the hydroxyl-radical-based self-cleansing of the troposphere, Nature Geosci, 7(8), 559–563, doi:10.1038/NGEO2199, 2014.

Logan, J. A., Prather, M. J., Wofsy, S. C. and McElroy, M. B.: Tropospheric chemistry: A global perspective, J. Geophys. Res., 86(C8), 7210–7254, doi:10.1029/JC086iC08p07210, 1981.

Mao, J., Ren, X., Zhang, L., Van Duin, D. M., Cohen, R. C., Park, J. H., Goldstein, A. H., Paulot, F., Beaver, M. R., Crounse, J. D., Wennberg, P. O., Digangi, J. P.,

Henry, S. B., Keutsch, F. N., Park, C., Schade, G. W., Wolfe, G. M., Thornton, J. A. and Brune, W. H.: Insights into hydroxyl measurements and atmospheric oxidation in a California forest, Atmos. Chem. Phys., 12(17), 8009–8020, doi:10.5194/acp-12-8009-2012, 2012.

Overall, this paper should be acceptable for ACP subject to revisions somewhere between minor and major.

Details

Abstract: Maybe the increase of oxidation capacity dowwind a BB event can be more clearly put into the abstract ?

Line 29ff : Edit the sentence 'The oxidation of isoprene is higher....'

Line 70ff: It strikes me that this part does not make any reference to the important Caltech studies starting with Paulot et al.

Line 146 ff: How do the authors assure that ISOPOOH does contribute to the measured m/z = 73 at all ? Has there been any test for this ? How is the inlet system designed to allow measurement of this species ?

We reported the data at m/z 71 as the sum of 3 isomers. According to Rivera-Rios et al., 2014, the conversion yields of ISOPOOH into MVK and MACR was observed to be greater than 70%, but the decomposition is known to be highly sensitive to instrumental settings such as temperature, contact time and type of surface materials, especially transition metal surfaces (Liu et al., 2013; Nguyen et al., 2014; Rivera-Rios et al., 2014, Liu et al., 2016, Bernhammer et al., 2017).

Liu, Y. J., Herdlinger-Blatt, I., McKinney, K. A. and Martin, S. T.: Production of methyl vinyl ketone and methacrolein via the hydroperoxyl pathway of isoprene oxidation, Atmos. Chem. Phys., 13(11), 5715–5730, doi:10.5194/acp-13-5715-2013, 2013.

Nguyen, T. B., Crounse, J. D., Schwantes, R. H., Teng, A. P., Bates, K. H., Zhang, X., St Clair, J. M., Brune, W. H., Tyndall, G. S., Keutsch, F. N., Seinfeld, J. H. and Wennberg, P. O.: Overview of the Focused Isoprene eXperiment at the California Institute of Technology (FIXCIT): mechanistic chamber studies on the oxidation of biogenic compounds, Atmos. Chem. Phys., 14(24), 13531–13549, doi:10.5194/acp-14-13531-2014, 2014.

Rivera-Rios, J. C., Nguyen, T. B., Crounse, J. D., Jud, W., Clair, J. M. S., Mikoviny, T., Gilman, J. B., Lerner, B. M., Kaiser, J. B., Gouw, J., Wisthaler, A., Hansel, A., Wennberg, P. O., Seinfeld, J. H. and Keutsch, F. N.: Conversion of hydroperoxides to carbonyls in field and laboratory instrumentation: Observational bias in diagnosing pristine versus anthropogenically controlled atmospheric chemistry,, 1–7, doi:10.1002/(ISSN)1944-8007, 2014.

Liu, Y., Brito, J., Dorris, M. R., Rivera-Rios, J. C., Seco, R., Bates, K. H., Artaxo, P., Duvoisin, S., Keutsch, F. N., Kim, S., Goldstein, A. H., Guenther, A. B., Manzi, A. O., Souza, R. A. F., Springston, S. R., Watson, T. B., McKinney, K. A. and Martin, S. T.: Isoprene photochemistry over the Amazon rainforest, Proc. Natl. Acad. Sci. U.S.A., 113(22), 6125–6130, doi:10.1073/pnas.1524136113, 2016.

Bernhammer, A.-K., Breitenlechner, M., Keutsch, F. N. and Hansel, A.: Technical note: Conversion of isoprene hydroxy hydroperoxides (ISOPOOHs) on metal environmental simulation chamber walls, Atmos. Chem. Phys., 17(6), 4053–4062, doi:10.5194/acp-17-4053-2017, 2017.

Line 200 ff: Isn't such quasi-analytical approach prone to errors ? How do OH levels calculated by eqn (2) compare to detailed model results ? Can this be compared to establish the validity of eqn (2) ? How uncertain are the OH concentrations calculated ? Is there any chance to compare the analytical results with, say, a 1-D or a box-trajectory model result ?

We appreciate your question and we agree that more tests are necessary. We

added the standard error in the Table 4 and in Figure 9 the intervals represent the level of confidence (0.95) used. Your question is very important and guided us to insist in the development of the equation. After improvements and suggestions made by the Referees, we expect in the future apply the indirect [OH] calculation in atmospheric models as a diagnostic tool. We are aware that a lot of work must be done, but still believe that is possible to have it such as tool.

Line 203: ISOPOOH , not ISOPOHH

Line 363: Maybe I oversee something but what exactly is that 'Sequential reaction approach' ?

Sequential reaction approach or simple consecutive reaction scheme model is an expression for the time rate of change in the [MVK+MACR+ ISOPOOH]/[Isoprene] ratio and derived as a function of [OH], the rate coefficients, and the time available for processing. A consecutive reaction scheme, in which isoprene and the reaction products MVK, MACR and ISOPOOH react with OH is shown by equation (2).

**Biomass burning emissions disturbances on the isoprene oxidation in a tropical forest**

Fernando C. Santos[1], Karla M. Longo[2], Alex B. Guenther[3], Saewung Kim[3], Dasa Gu[3], Dave E. Oram[4], Grant L. Forster[4], James Lee[5], James R. Hopkins[5], Joel Brito[6*] and Saulo R. Freitas[7]

[1] Earth System Science Center, National Institute for Space Research, São José dos Campos, SP, Brazil
[2] Universities Space Research Association/Goddard Earth Sciences Technology and Research, NASA Goddard Space Flight Center, Greenbelt, MD, USA
[3] Department of Earth System Science, University of California, Irvine, CA, USA
[4] National Centre for Atmospheric Science, School of Environmental Sciences, University of East Anglia, Norwich, UK
[5] National Centre for Atmospheric Science, Department of Chemistry, University of York, York, UK
[6] University of São Paulo, São Paulo, SP, Brazil
[7] Universities Space Research Association/Goddard Earth Sciences Technology and Research, NASA Goddard Space Flight Center, Greenbelt, USA

*Correspondence to*: Fernando C. dos Santos (fcsantos@if.usp.br)

[1]*Now at: Instituto de Física, Universidade de São Paulo, São Paulo, Brasil

[revised manuscript text omitted]